# Freezing solute atoms in nanograined aluminum alloys via high-density vacancies

Shenghua Wu [1], Hanne S. Soreide[2], Bin Chen[3], Jianjun Bian[4], Chong Yang[1], Chunan Li[2], Peng Zhang[1], Pengming Cheng[1], Jinyu Zhang[1], Yong Peng[3], Gang Liu [1✉], Yanjun Li [2✉], Hans J. Roven [2✉] & Jun Sun [1✉]

Low-temperature decomposition of supersaturated solid solution into unfavorable intergranular precipitates is a long-standing bottleneck limiting the practical applications of nanograined aluminum alloys that are prepared by severe plastic deformation. Minimizing the vacancy concentration is generally regarded as an effective approach in suppressing the decomposition process. Here we report a counterintuitive strategy to stabilize supersaturated solid solution in nanograined Al-Cu alloys via high-density vacancies in combination with Sc microalloying. By generating a two orders of magnitude higher concentration of vacancies bonded in strong (Cu, Sc, vacancy)-rich atomic complexes, a high thermal stability is achieved in an Al-Cu-Sc alloy that precipitation is nearly suppressed up to ~230 °C. The solute-vacancy complexes also enable the nanograined Al-Cu alloys with higher strength, greater strain hardening capability and ductility. These findings provide perspectives towards the great potentials of solute-vacancy interaction and the development of nanograined alloys with high stability and well-performed mechanical properties.

[1] State Key Laboratory for Mechanical Behavior of Materials, School of Materials Science and Engineering, Xi'an Jiaotong University, Xi'an 710049, China. [2] Department of Materials Science and Engineering, Norwegian University of Science and Technology, 7491 Trondheim, Norway. [3] Key Laboratory of Magnetism and Magnetic Materials of the Ministry of Education, School of Physical Science and Technology and Electron Microscopy Centre of Lanzhou University, Lanzhou University, Lanzhou 730000, China. [4] Department of Industrial Engineering, University of Padova, Via Gradenigo 6/a, Padua 35131, Italy. ✉email: lgsammer@xjtu.edu.cn; yanjun.li@ntnu.no; hans.j.roven@ntnu.no; junsun@xjtu.edu.cn

As an important phase transformation highly focused on metal materials, solid-state precipitation enables microstructural tuneability at various length scales and property optimization upon different demands[1,2]. Research on the solid-state precipitation over the past several decades has followed a trajectory of artificial controlling, as well demonstrated in both structural alloys (e.g., high-strength aluminum alloys[3], copper alloys[4], and steels[5]) and functional materials (e.g., shape-memory alloys[6], magnets[7], and thermoelectrics[8]). It has been generally recognized that the precipitation kinetics is dominated by atomic diffusion[9,10], where vacancies play a critical role specially for substitutional alloying elements[11]. The artificial controlling of precipitation thus could be advanced through the in-depth understanding of interactions between vacancies and solute atoms. A typical example is to use the microalloying effect in heat-treatable aluminum (Al) alloys to adjust the precipitation behaviors. Minor addition of In, Sn, or Cd into Al-Cu alloys was found to suppress natural aging while promoting precipitation at elevated temperatures[12]. The suppression of natural aging is associated with a strong binding between the microalloying element (In, Sn, or Cd) and vacancy. Such a strong binding traps the quenched vacancies effectively and hence slows down the Cu diffusion greatly[12]. But the vacancies are released at elevated temperatures which facilitate the precipitation of $\theta'$-$Al_2Cu$ precipitates. Similar precipitation behaviors with the same mechanisms were also observed in Al-Mg-Si alloys microalloyed by Sn[13]. Recently, the vacancy-demanded precipitation was directly confirmed in delicately-designed low dimension material geometries, where the vacancies were either highly boosted in numbers[14] (vacancies stimulated at the surface by heating) or fully eliminated by diffusion[15] (vacancies annihilated at the surface by thinning), leading to promoted or suppressed precipitation, respectively, in small-sized samples. All the previous results exclusively directed to the same conclusion that excess vacancies are necessary to promote the precipitation in Al alloys.

Severe plastic deformation (SPD) (e.g., high-pressure torsion (HPT), and equal-channel angular pressing (ECAP)) has been widely applied to generate high-strength bulk Al alloys of submicron and nanosized grain structures for potential applications[16,17]. High content of solute elements is critical for the alloys to reach nanograined (NG) structure by retarding the recovery and increase the strength by solution hardening. However, the high strain applied during SPD inevitably produces high-density crystal defects in the small-grained Al alloys, including non-equilibrium grain boundaries, dislocations, and vacancies[16,18]. In particular, the vacancy concentration can typically reach a level of $\sim 10^{-3}$ at.% in the as-processed metal samples by HPT[19], at least one order of magnitude greater than quenched-in vacancies in samples from conventional solution treatments[20,21]. These super-excess crystal defects greatly accelerate the atomic diffusion and concomitantly trigger precipitations at lower temperatures, preferentially along dislocations and grain boundaries[16]. In SPD-processed Al-Cu alloys with nanosized grains[20,22], for instance, a large quantity of intergranular incoherent stable $\theta$-$Al_2Cu$ phase could form at grain boundaries (GBs) even during ambient storage. The truncated precipitation sequence bypasses the intragranular precipitations of metastable coherent phases $\theta''$ and $\theta'$, normal for the artificial aging of coarse-grained counterparts. Such a catastrophic precipitation behavior greatly reduces the strengthening potential by artificial aging of NG alloys produced by SPD processing[16]. Another consequence of such decomposition of supersaturated solid solution is the significant reduction in strength at elevated temperature, due to the rapid recovery and grain coarsening[23]. The intractable low-temperature (generally below $\sim 100$ °C and even at room temperature) precipitation of stable precipitate phases

becomes another challenge of thermal instability that seriously limits the practical usage of NG Al alloys and other NG alloys with supersaturated solid solution[16] at elevated temperatures, in parallel with the widely-concerned severe grain coarsening[24].

Minimizing crystal defects is intuitively a strategy to slow down atomic diffusion and avoid unfavorable low-temperature precipitation in the NG alloys. This strategy was recently manifested in NG supersaturated Al-Mg alloys[23], where an average grain size ($d$) of $\sim 8$ nm was achieved by HPT processing at 77 K. A Schwarz crystal structure with zero mean curvature constrained by twin boundaries was unprecedently produced[23]. Due to the ultrafine nano grain size, the vacancy concentration is quite low within the grains[23,25]. As a result, the diffusion-controlled $Al_3Mg_2$ precipitation from supersaturated nanograins is completely suppressed at temperatures as high as 450°C. In contrast, in the NG Al-Mg alloys with an average grain size of $\sim 50$ nm, intergranular $Al_3Mg_2$ precipitation at low temperatures was evident, inspired by the excessive crystal defects[23].

Here we report an inverse strategy to stabilize the supersaturated solute solution, suppressing the unfavorable precipitation in NG Al-Cu alloys. This strategy is counterintuitively to increase the vacancy concentration to a substantially higher level and utilize microalloying elements (scandium, Sc) to produce strong solute-vacancy complexes. We demonstrate that these (Cu, Sc, vacancy)-rich atomic complexes firmly trap the vacancies and substantially suppress precipitation of Al-Cu precipitates up to $\sim 230$ °C. The high-density (Cu, Sc, vacancy)-rich atomic complexes also enable the NG Al-Cu alloys to simultaneously possess higher strength, greater work hardening, and larger ductility. The solute-vacancy complex design concept, not limited by the ultrafine nano grain size and readily applicable in the engineering of large-sized samples, offers a different pathway to develop NG Al alloys or other NG structured alloys with controllable solid-state precipitations and a good strength/ductility combination.

## Results

**High vacancy concentration achieved by 77 K-HPT**. Figure 1a shows a representative transmission electron microscopy (TEM) image of the as-processed Al-2.5 wt.%Cu-0.3 wt.%Sc alloy by HPT at liquid nitrogen/cryogenic temperature (hereafter denoted as the AlCuSc-C alloy). Equiaxial nanosized grains with random orientation are evident in this alloy. The size of grains is rather uniform, with an average grain size $d$ of $\sim 100$ nm. The element distributions across the grains are also homogeneous, showing no apparent element enrichment or depletion regions (Fig. 1b). For comparison, three other alloys were prepared (see Methods), including the Sc-free Al-2.5 wt.%Cu alloy similarly by the 77 K-HPT (denoted as the AlCu-C alloy), and the Al-2.5 wt.%Cu alloys with and without 0.3 wt.%Sc addition by traditional room temperature (298 K) HPT (denoted as the AlCuSc-R and AlCu-R alloy, correspondingly). All the three alloys had NG equiaxial grains, but with different grain sizes, i.e., the AlCu-R alloy of $\sim 150$ nm, AlCuSc-R alloy of $\sim 135$ nm, and AlCu-C alloy of $\sim 112$ nm (see Supplementary Fig. 1), indicating that both the cryogenic deformation and the Sc microalloying can enhance the grain refinement during HPT. The AlCuSc-C alloy also shows a higher proportion of low angle grain boundaries (LAGBs) and dislocation density than the rest of the alloys (Supplementary Fig. 2 and Fig. 3). The main reason is that the dislocation migration and therefore the dynamic recovery during HPT processing could be effectively inhibited by low temperature as well as by the Zener pinning from Sc atoms[26].

The most unexpected difference among the four alloys was lying in the vacancy concentration. This is evidenced by the huge

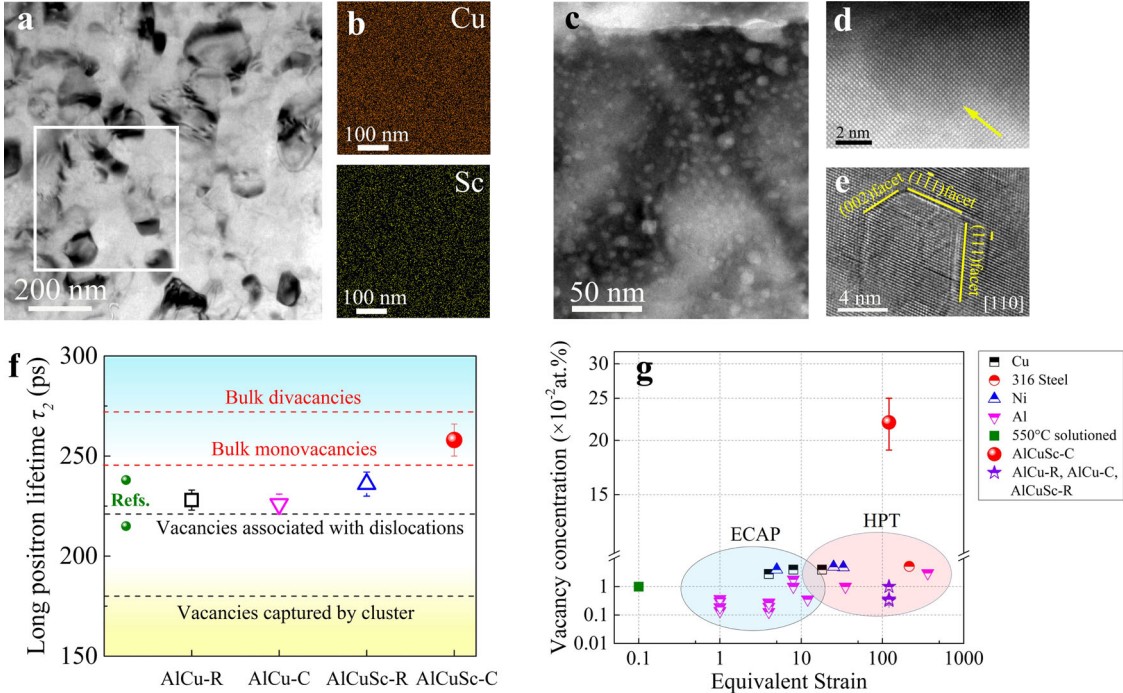

**Fig. 1 A high vacancy concentration in the AlCuSc-C alloy. a** Typical bright-field TEM image showing the grains in the AlCuSc-C alloy. **b** EDS elemental mappings of Cu and Sc elements in the area marked by the rectangular framework in **a**. **c** Typical bright-field TEM image of the AlCuSc-C alloy after ion milling at low energy (3.1 kV) and low incidental angle (4°) for 20 min. **d** Typical atomic-resolution HAADF-STEM image viewed along <100>$_{Al}$ showing a void. **e** High-resolution TEM image viewed along <110>$_{Al}$ showing the void. **f** Measured positron annihilation lifetime of the AlCu-R, AlCu-C, AlCuSc-R, and AlCuSc-C alloys, compared with typical values of room temperature HPT-processed Al alloys in refs. [19,30]. The error bars represent standard deviations from the mean for sets of three tests. **g** A comparison of vacancy concentrations between the AlCu-R, AlCu-C, AlCuSc-R, AlCuSc-C alloys, and other SPD-processed alloys, including Cu[34], 316 steels[35], Ni[34,36], and Al alloys[18,31]. The error bar on the red data point represents standard deviations from the mean for sets of three tests.

number of nanosized "voids" homogeneously dispersed within the TEM samples of the AlCuSc-C alloy (see Fig. 1c) prepared purposely by the ion milling method. Such nanosized voids could not be observed in the other three alloys. The nanosized voids in the AlCuSc-C alloy have visible facets (Fig. 1d), showing a truncated octahedron shape viewed along the <110>$_{Al}$ direction (Fig. 1e), which are similar to the voids in pure Al grown from high temperature-induced vacancies[27]. Additional evidence can be seen in Supplementary Fig. 4. These nanosized voids were not inherently existed in the as-prepared AlCuSc-C alloy, they were created through the coalescence of original vacancies, triggered by low-energy and low-angle ion milling. Under Ar ion bombardment, the collision cascades and induced temperature increase cause the aggregation of vacancies into voids[28,29]. This implies that a substantially higher concentration of vacancies had been achieved in the NG AlCuSc-C alloy than in the other three NG alloys, despite their comparable grain structure and dislocation density.

The positron annihilation technique was applied to measure vacancy density in these alloys (see Methods). The experimental results of positron lifetime, as shown in Fig. 1f, highlight the difference. The average lifetime of the as-processed NG AlCu-R, AlCu-C, and AlCuSc-R alloys is located within the range from 226 to 236 ps, close to typical values of the HPT-processed NG Al alloys[19,30]. While the as-processed NG AlCuSc-C alloy stands out, with an average lifetime of ~ 258 ps larger than the above range. It's generally recognized that the critical lifetime of vacancies associated with dislocations, bulk monovacancies, and bulk divacancies corresponds to ~220, 245, and 273 ps in Al, respectively[19,31]. Besides, the lifetime of vacancies trapped in copper clusters in Al was reported to be ~180 ps[32]. Based on the experimental results in Fig. 1f and Supplementary Table 1, the

average lifetimes determined in the NG AlCu-R, AlCu-C, and AlCuSc-R alloys are mainly due to the positron annihilation in vacancies associated with dislocations and bulk monovacancies. However, the measured lifetime in the NG AlCuSc-C alloy is suggested to be dominated by the positron annihilation in bulk monovacancies and bulk divacancies[31].

By applying the positron standard trapping models and diffusion trapping models[31,33], it was calculated that a substantially higher vacancy concentration value ($C_v$) of ~$22 \times 10^{-2}$ at.% has been achieved in the as-processed NG AlCuSc-C alloy. This high vacancy concentration is one to two orders of magnitude higher than that in the present NG AlCu-R, AlCu-C, and AlCuSc-R alloys (all <$1 \times 10^{-2}$ at.%), as well as other small-grained alloys processed by HPT or ECAP (0.1–$2 \times 10^{-2}$ at.%) (Fig. 1g)[18,31,34–36].

In fact, the production rate of vacancies during HPT was estimated as ~$10^{-5}\,s^{-1}$ at room temperature in Al-5.8 wt.%Mg alloy[37]. Since a great number of non-equilibrium grain boundaries and abundant dislocations will act as efficient sinks to annihilate vacancies[38], the HPT samples after processing for 600 s typically have a low vacancy concentration of only ~ $0.1 \times 10^{-2}$ at.% (about 1/600 of the theoretical one). While in the present NG AlCuSc-C alloy processed by HPT for 600 s, over a third of the vacancies produced survived, leading to >30% of the theoretical vacancy concentration retained. In comparison, the vacancy concentration $C_v$ is just slightly elevated from ~$0.3 \times 10^{-2}$ at.% in the AlCu-R alloy to ~$0.4 \times 10^{-2}$ at.% in the AlCu-C alloy, and to ~$1.0 \times 10^{-2}$ at.% in the AlCuSc-R alloy. The individual effect of 77 K-HPT or Sc microalloying appears weak in promoting $C_v$. A coupling between the two effects is so strong that boosts the vacancy concentration to a significantly high level (~$22 \times 10^{-2}$ at.%) in the AlCuSc-C alloy.

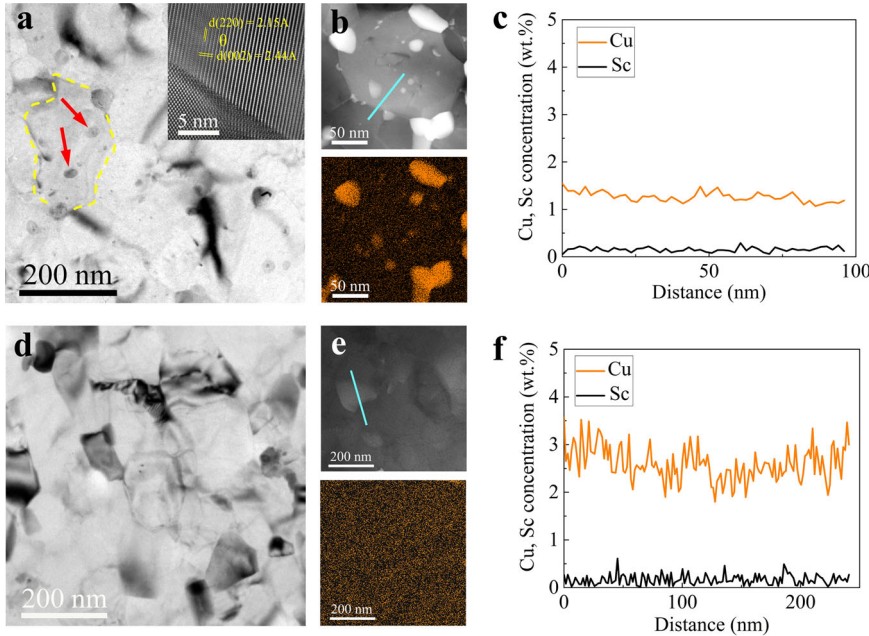

**Fig. 2 Microstructural characterization of the AlCuSc-R and AlCuSc-C alloys aged at 125 °C for 6 h. a** Typical bright-field TEM image of the AlCuSc-R alloy (inset is a high-resolution TEM image of the $\theta$ phase). **b** Typical HAADF-STEM image of the AlCuSc-R alloy and corresponding elemental mapping of Cu. **c** EDS line scanning result corresponding to the blue line marked in **b**. **d** Typical bright-field TEM image of the AlCuSc-C alloy. **e** Typical HAADF-STEM image of the AlCuSc-C alloy and corresponding elemental mapping of Cu. **f** EDS line scanning result corresponding to the blue line marked in **e**.

**High thermal stability and strong solute-vacancy complexes**. Artificial aging at 125 °C has been generally applied to investigate precipitation behaviors in the SPD-processed Al alloys[18,22]. Here we exposed the NG alloys to 125 °C-aging and compared their thermal stability. In the AlCuSc-R alloy aged for 6 h, substantial precipitation of equilibrium $\theta$-Al$_2$Cu phase along GBs and within the grain interior has occurred (Fig. 2a). The Cu-rich regions dispersed in Fig. 2b correspond to the coarse $\theta$ particles. The Cu atoms that survived in the matrix have a concentration of only ~1.2 wt.% (Fig. 2c), about 50% of the nominal Cu content (~2.5 wt.%). Similar precipitation of coarse stable $\theta$ precipitates also occurred in the AlCu-R and AlCu-C alloys during artificial aging at 125 °C (Supplementary Fig. 5), and even during long-time room temperature storage (Supplementary Fig. 6). In contrast, no precipitates could be detected after aging for 6 h in the AlCuSc-C alloy (Fig. 2d). The distribution of Cu atoms was as homogeneous as that in the as-processed state (Figs. 2e, f, 1b). It is evident that the NG AlCuSc-C alloy displayed high thermal stability against the unfavorable precipitation of coarse stable $\theta$ precipitates. The high thermal stability is also manifested in hardness measurement. The AlCuSc-C alloy maintained a peak hardness plateau up to 100 h when aged at 125 °C, but the AlCu-R and AlCuSc-R alloys showed an apparent hardness drop after aging for 6 h (Supplementary Fig. 7).

To reveal the precipitation kinetics of the alloys, DSC experiments were performed within a wide temperature range. Figure 3a shows representative DSC curves of the as-processed NG AlCuSc-R and AlCuSc-C alloys. Also presented in Fig. 3a for comparison is the DSC curve of a coarse-grained Al-2.5 wt.%Cu-0.3 wt.%Sc alloy after solution treatment and water quenching. The coarse-grained Al-Cu-Sc alloy displayed a typical heat flow curve with the exothermic peak happened at ~150 °C representing an intragranular $\theta'$-Al$_2$Cu precipitation. In the AlCuSc-R alloy with nanosized grains, the $\theta$ precipitation was highly accelerated (see Fig. 2a) and the exothermic peak was shifted to a lower temperature of ~125 °C. The subsequent endotherm reaction (from 125 to 400 °C) should be mainly associated with the coarsening and partial dissolution of the $\theta$

precipitates[39]. In contrast, no obvious exothermic peak can be observed in the NG AlCuSc-C alloy up to 200 °C. Thereafter, the exothermic heat flow emerged and climbed up gradually until about 400 °C. The DSC results clearly illustrate that the precipitation in the AlCuSc-C alloy was much delayed at higher temperatures (above ~200 °C) and very sluggish at higher temperatures from ~200 to ~400 °C. This agrees well with the TEM observation in Fig. 3a. The long-standing problem of low-temperature unfavorable precipitation generally existed in the NG Al alloys (including in present AlCu-R, AlCu-C, and AlCuSc-R alloys) is dexterously ironed out in the AlCuSc-C alloy, indicative of a significant improvement in the thermal stability.

We have made a statistic about the thermal instability temperature of some Al alloys[13,20,22,40–46] as a function of deformation strain (quantified here by equivalent strain, $\varepsilon_{eq}$), as illustrated in Fig. 3b. The observed critical temperature ($T_i$) refers to the aging temperature at which appreciable precipitation of stable precipitates could be detected within 50 h of heat treatment. As shown, $T_i$ sharply decreases with increasing deformation strain experienced by the alloys. The alloys undergo greater deformation strain are expected to have smaller grains and more crystal defects[16], which strongly stimulate earlier precipitation of stable precipitate phases and concomitantly lead to a lower $T_i$. These results assume a general trend of $T_i$ reducing with raising $\varepsilon_{eq}$, and vice versa. While in this work, the balancing dilemma between $T_i$ and $\varepsilon_{eq}$ was evaded in the AlCuSc-C alloy that exhibited a high $T_i$ of ~230 °C after a severe HPT processing with $\varepsilon_{eq}$ ~120. The precipitation kinetics of the NG AlCuSc-C alloy is even more sluggish than the solution-treated coarse-grained counterpart. Of special interest to note is that, in the AlCuSc-C alloy aged at ~225 °C for 50 h, a significant grain coarsening has occurred (Fig. 3c). However, the Cu atoms were still kept in a homogeneous distribution in the matrix (Fig. 3d). Intergranular precipitation of globular-shaped $\theta$ particles is still rare.

To reveal the element distribution of Cu and Sc atoms in the NG AlCuSc-C alloy, samples after both natural aging for 6 months

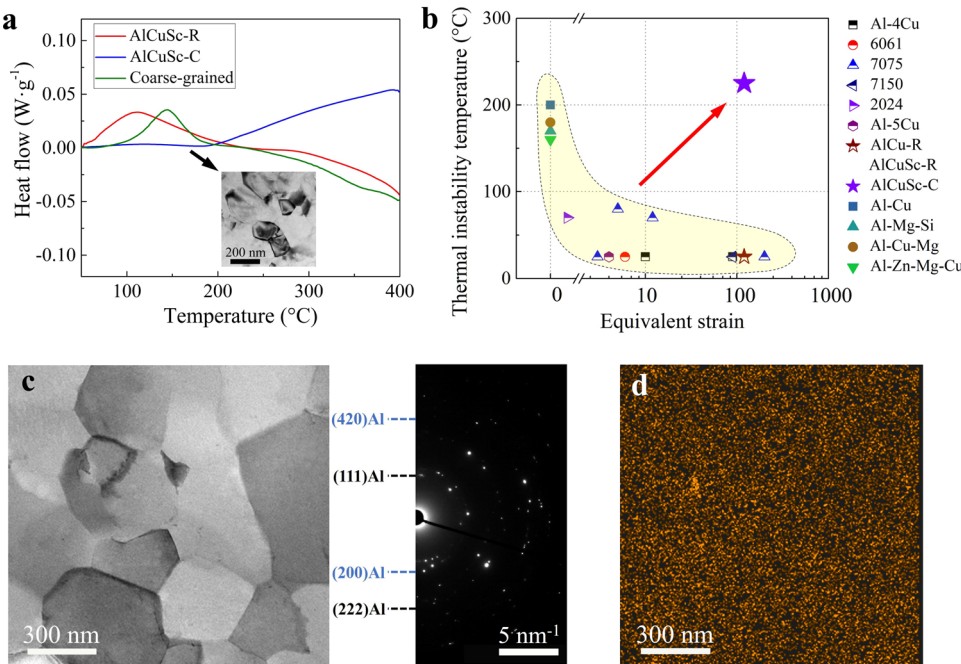

**Fig. 3 High thermal stability of the AlCuSc-C alloy. a** DSC heating curves of the AlCuSc-R, AlCuSc-C alloys, and coarse-grained Al-Cu-Sc alloy, inset is a typical bright-field TEM image of AlCuSc-C aged at 175 °C for 50 h. **b** The minimum temperature for thermal instability of some typical cast Al alloys, including Al-Cu[44], Al-Mg-Si[13], Al-Cu-Mg[46], Al-Zn-Mg-Cu[45], and SPD-processed Al alloys, such as Al-4Cu[20], 6061[40], 7075[41], 7150[42], 2024[43], and Al-5Cu[22]. **c** Typical bright-field TEM image and corresponding selected area electron diffraction (SAED) pattern of the AlCuSc-C alloy aged at 225 °C for 50 h. **d** Corresponding elemental mapping of Cu in **c**.

and 125 °C-artificial aging for 6 h, have been characterized by atom probe tomography (APT)[47]. No precipitates or solute clusters (see Methods) can be found in the sample after cryogenic HPT (Fig. 4a), confirming that no precipitation has happened during natural aging. The nearest neighbor distance distribution profiles of Cu and Sc ions (Supplementary Fig. 8) show that the Cu and Sc atoms are distributed nearly randomly within the grains. Interestingly, discernible segregation of Cu atoms as well as slight segregation of Sc atoms at GBs can be evidenced by the concentration profile across one GB (Fig. 4c). A clear solute depletion zone with a width of ~5 nm at either side of the GB can be observed. The average Cu concentration in the grain and in the GB depletion zone is measured as ~1.05 at.% (or ~2.5 wt.%) and ~0.15 at.% (or ~0.35 wt.%), respectively. The enrichment of Cu atoms at GBs is obviously a result of the diffusion of Cu and Sc atoms from the matrix to the GB. But the diffusion zone is limited to a very thin region adjacent to the GB.

After 6h-artificial aging at 125 °C, no precipitates are observed in the intragranular region by APT. Occasionally, plate-shaped Cu atom clusters with a maximum Cu concentration of ~5.0 at.% can be observed (as marked by the dashed ellipse in Fig. 4d), which could be GP-I zones of Al-Cu precipitates. The GB concentration of Cu is somewhat higher while the intragranular Cu concentration (~0.86 at.%) is slightly lower than the as-HPT sample (Fig. 4c). This means that only a slight diffusion of Cu atoms from the matrix to the GB has happened during artificial ageing. Even though, no Cu-rich precipitates could be detected at GBs, indicating that the Cu atoms at GBs are rather stable, remaining in the solid solution state. Since there is not any precipitation of Al-Cu precipitates, neither in the grain nor at GBs, whereas the supersaturation of Cu in solid solution of grains is somewhat reduced, the significant increase of the hardness by artificial aging should be ascribed to the GB segregation strengthening by Cu atoms[22]. But this still cannot explain why there is no precipitation during natural and artificial aging.

Partial radial distribution function (RDF) analysis was used to further analyze the distribution of Cu and Sc atoms in the samples[48]. The normalized partial RDF around Sc atoms measured in the matrix of both samples are shown in Fig. 4b and e. As shown, in the naturally-aged sample, there is a notably higher Cu concentration within a radial distance of ~1 nm around the Sc atoms (Supplementary Fig. 9), implying that a strong Sc-Cu clustering exists in the matrix. A closer analysis of the partial RDF shows that Cu atoms are preferentially located at the first nearest neighbor (1NN) positions of Sc atoms, with a normalized RDF value approaching 2, where unity indicates a random distribution[48] (see Methods). At the second NN (2NN) positions of Sc atoms, the normalized RDF value is close to unity. After artificial aging, Cu atoms also show a higher concentration at 1NN positions of Sc atoms. However, the normalized RDF value is significantly lower than that before artificial aging. The above RDF analysis suggests that short-range ordering exists between Sc and Cu atoms at 1NN positions to each other in the AlCuSc-C alloy even after artificial aging. This implies that atom complexes containing locally-ordered Cu-Sc atoms have formed, in which the concentration of Cu and Sc atoms is not high enough to be visible as atom clusters in APT. To confirm the existence of atomic complexes enriched with Sc and Cu, high-angle annular dark-field STEM (HAADF-STEM) has been applied in the AlCuSc-C alloy. As shown in Fig. 4g–i, a high-density nanosized (~1 nm with a volumetric number density of ~$2.0 \times 10^{24}$ m$^{-3}$) features with significantly higher contrast than the surrounding matrix exists in the naturally-aged AlCuSc-C alloy. An EDX mapping of the complexes of bright atoms indicates that the complexes are enriched with Cu atoms, where a slight enrichment of Sc atoms can also be detected (Supplementary Fig. 10). The fast Fourier transform (FFT) pattern (Inset in Fig. 4g) reveals that weak reflections exist at the {110}$_{Al}$ positions, implying that the complexes have some short-range ordering. However, additional evidence shown in Supplementary Fig. 11 reveals that the atomic

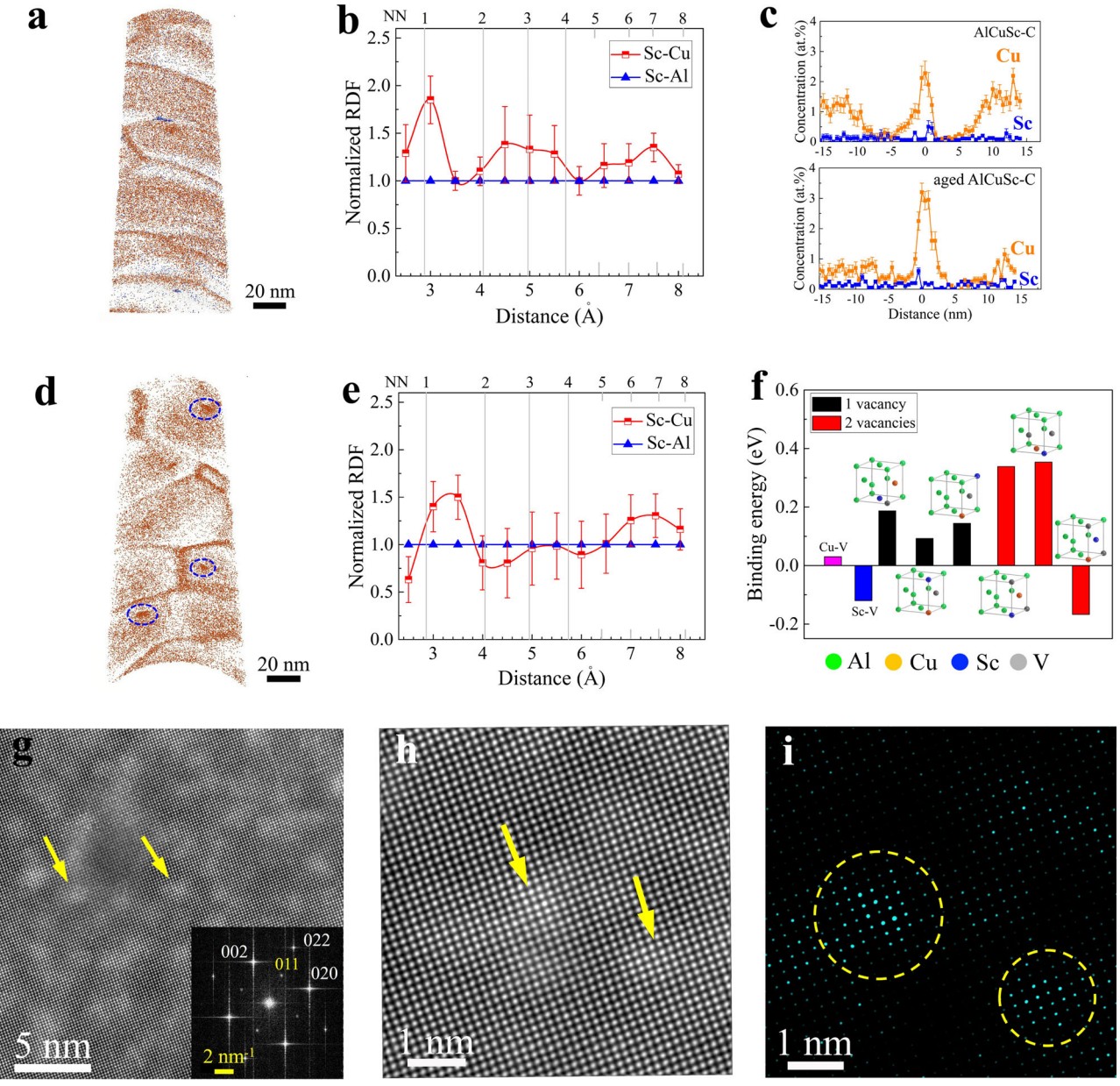

**Fig. 4 Solute complexes in the AlCuSc-C alloy. a** Representative APT reconstruction of the naturally-aged AlCuSc-C alloy, where Cu and Sc atoms are labeled with brown and blue colors, respectively. **b** Partial radial distribution function (RDF) analysis of the naturally-aged AlCuSc-C alloy showing the short-range ordering between Sc and Cu at the first nearest neighbor (NN) location. The error bars are standard deviations of the mean. **c** 1-D concentration profile of Cu across GBs in the AlCuSc-C alloy after natural aging and artificial aging at 125 °C for 6 h. The error bars are standard deviations of the mean. **d** Representative APT reconstruction of the AlCuSc-C alloy aged at 125 °C for 6 h, where plate-shaped Cu atom clusters are marked by a dashed ellipse. **e** RDF analysis of the AlCuSc-C alloy aged at 125 °C for 6 h. The error bars are standard deviations of the mean. **f** Binding energy of the Cu-Sc-X vacancy complex (X = 1, 2) calculated by DFT. **g** High-resolution HAADF-STEM image viewed along <100>$_{Al}$ of the AlCuSc-C alloy, inset is the corresponding fast Fourier transform (FFT) pattern. **h** Atomic-resolution <100>$_{Al}$ HAADF-STEM image showing the solute complexes in the AlCuSc-C alloy. **i** Inverse FFT image showing the atom complexes enriched with Cu and Sc.

complexes don't possess an L1$_2$ structure. In contrast, no apparent solute clusters can be observed in the AlCuSc-C alloy.

The formation of Cu-Sc atomic complexes in the AlCuSc-C alloy seems to contradict the previous density functional theory (DFT) calculation results[49], which show that the formation of Sc-Cu dimers is energetically unfavorable at either 1NN or 2NN positions in Al solid solution. It is known that by binding together with vacancies, the bonds of solute-solute atoms can be strengthened. Even though, only weak binding energy (max. ~0.1 eV) exists between Cu and Sc atoms in Cu-Sc-vacancy complexes[49], which may not explain the strong Sc-Cu short-range

ordering in the present work. Since the major difference between AlCuSc-C and AlCuSc-R alloys is that the former has a high concentration of vacancies, the completely different precipitation behaviors of the alloys may be attributed to the vacancy. We have done a systematic DFT calculation to assess the binding energy between Cu and Sc atoms when combined with two vacancies, namely in V-Sc-Cu-V (V denotes vacancy) complexes of different atomic arrangements. Surprisingly, a significantly higher binding energy can be obtained in a majority of the complexes with Cu and Sc atoms located at 1NN position to each other, and with a maximum binding energy of ~0.35 eV (Fig. 4f). It means that

when a Cu atom and a Sc atom are combined together with more than one vacancy, the complex has lower free energy and is thermodynamically more stable. This explains why strong Sc-Cu short-range ordering could exist in the AlCuSc-C alloy, as revealed by the APT characterization and HAADF-STEM. Further analysis of the DFT calculation results reveals that the enhanced binding energy with vacancy involved originates physically from the increased charge density (see Supplementary Fig. 12). In order to further verify that Sc atoms are located next to the vacancies, coincidence Doppler broadening (CDB) experiments[50,51] were additionally performed to obtain the CDB ratio curves for pure Cu, pure Sc, and the AlCuSc-C alloy (see Methods and Supplementary Fig. 13). The CDB ratio curve of the AlCuSc-C alloy shows a mixture of the characteristic Cu signal and Sc signal: not only a long tail in the high momentum region ($>15 \times 10^{-3}$ $m_0c$) due to Cu electrons but also a hump around $10 \times 10^{-3}$ $m_0c$ due to Sc electrons. The agreements indicate that a large fraction of positrons annihilates at vacancies located next to Sc atoms[50,51].

In Al alloys, interactions between solute atoms and vacancies can assist the formation of solute clusters and precipitates through the diffusion kinetics process[10,52,53]. The low-temperature (even room temperature) unfavorable precipitation generally observed in the NG Al alloys is mainly due to the high-density vacancies and other defects (dislocation and GBs) introduced by the severe deformation, which provide effective diffusion paths for solute atoms and nucleation sites of a low-energy barrier for precipitates[16,20]. During HPT of the Al-Cu-Sc alloy, the diffusivity of vacancies generated by deformation can be significantly reduced by the cryogenic temperature while the formation of V-Sc-Cu-V complexes can further stabilize the vacancies, as Cu atoms and especially Sc atoms have much lower diffusivities than vacancy. Sc atoms have a diffusivity several orders of magnitude lower than Cu atoms in the Al matrix[18,44]. Though we have only calculated the binding energy of V-Sc-Cu-V complexes, it can be expected that larger complexes containing more Cu and Sc atoms together with excess vacancies will also have high thermal stability and thus form, especially during room temperature storage, which are referred to as (Cu, Sc, vacancy)-rich atomic complexes herein. As a result, a high concentration of vacancies can be reserved in the AlCuSc-C alloy even after natural ageing for 6 months. It should be mentioned that a large part of bulk monovacancies and bulk divacancies exist in (Cu, Sc, vacancy)-rich atomic complexes. The positron annihilation lifetime in bulk monovacancies and bulk divacancies of the AlCuSc-C alloy arises mainly from the positron annihilation lifetime in such atomic complexes. On the other hand, since most of the vacancies are locked in the low-energy (Cu, Sc, vacancy)-rich atomic complexes, the diffusion-controlled precipitation of stable Al-Cu precipitates is significantly suppressed during room temperature storage. This is in agreement with a previous atomic-scale simulation work[54], which showed that when a solute atom is binding with two vacancies, the diffusivity of the solute will be significantly reduced. It is striking that intergranular precipitation of $\theta$ precipitates does not occur during artificial ageing at elevated temperatures.

It has already been demonstrated that the addition of Sc in coarse-grained Al-Cu alloys with quenched-in vacancies, can slow down the Cu diffusion and tailor the precipitation kinetics[44,55]. However, during room temperature HPT, the vacancies generated by deformation can be easily annihilated at sinking sources like dislocations and GBs due to the substantially higher diffusivity of vacancies. Therefore, there are not enough vacancies to form a high fraction of (Cu, Sc, vacancy)-rich atomic complexes containing a large fraction of vacancies in the AlCuSc-R alloy (see Supplementary Fig. 14). As a consequence, the Sc effect in

slowing down Cu diffusion is overwhelmed by the high-density dislocations and the unfavorable precipitation of stable $\theta$ precipitates can still be happening during room temperature storage. The formation of (Cu, Sc, vacancy)-rich atomic complexes in the AlCuSc-C alloy can reduce the concentration of free vacancies in the matrix and the diffusivity of Cu atoms, preventing the unfavorable $\theta$-precipitation during room temperature storage. This signifies a possibility to stabilize solute-vacancy complexes in a counterintuitive way, i.e., creating and maintaining vacancies in such a high concentration that allows more than one vacancy involved in the formation of solute-vacancy complexes. This scenario seems impossible in coarse-grained Al alloys prepared by conventional thermomechanical processes because the quenched-in vacancies are generally much less than the solute atoms[21]. But in the present AlCuSc-C alloy, the vacancy concentration was raised to a substantially high level (~0.22 at.%), which is about 1.6 times the Sc concentration (~0.14 at.%) in the supersaturated solid solution. A large proportion of the Cu-Sc-vacancy complexes might contain two or more vacancies. It is the Sc atom rather than the Cu atom that traps two or more vacancies to constitute (Cu, Sc, vacancy)-rich atomic complexes. The vacancies were almost entirely ensnared within the more stable complexes. As a result, the vacancy-assisted Cu diffusion for precipitation was nearly inhibited in the AlCuSc-C alloy.

**High strength and great strain hardening capability.** The tensile engineering stress-strain curves in Fig. 5a show that the AlCuSc-C alloy presents much better mechanical performance (tensile strength approaching ~570 MPa and uniform ductility approaching ~8.5%) in comparison with that in the AlCuSc-R alloy, coarse-grained Al-2.5 wt.%Cu and Al-2.5 wt.%Cu-0.3 wt.% Sc alloys in peak-aged conditions and with plate-like $\theta'$ precipitates[55]. We also have utilized a micropillar compressive test to further investigate the strain hardening capability and mechanical properties of the AlCuSc-C alloy, with the AlCuSc-R alloy as a reference. The micropillars used for testing have the same diameter of ~1 μm (about 100 grains enclosed in the cross-section). Representative engineering stress-strain curves of the samples subjected to different strain rates are shown in Fig. 5b. The AlCuSc-C alloy exhibits a much higher strain hardening capability than the AlCuSc-R alloy, at the same applied strain rate of $2 \times 10^{-4}$ $s^{-1}$. At 35% strain, the flow stress of the AlCuSc-C alloy is ~230 MPa higher than the AlCuSc-R alloy. The corresponding Kocks-Mecking plots inset in Fig. 5b clearly reveals that the AlCuSc-C alloy has a larger initial strain hardening rate and limited dynamic recovery (as characterized by the parameter $\beta$, which is defined as the slope marked by the dash-doted line). It means that the annihilation of dislocations is effectively inhibited in the AlCuSc-C alloy. The strain hardening rate as measured by $\Theta_{pillar} = \frac{\sigma_{5\%} - \sigma_{2\%}}{5\% - 2\%} n$ ($\sigma_{5\%}$ and $\sigma_{2\%}$ are the stress at the strain of 5 and 2%, respectively; and $n$ is the strain hardening exponent)[56] is determined to be ~3.9 GPa in the AlCuSc-C alloy (Fig. 5c), which is much greater than that of the AlCuSc-R alloy (~2.0 GPa) and reported values of NG pure Al (~0.15 GPa), quenched coarse-grained Al-2.5 wt.%Cu alloy (~0.19 GPa for <110>-oriented micropillar) and peak-aged coarse-grained Al-2.5 wt.%Cu alloy (~0.33 GPa for <110>-oriented micropillar) (see Supplementary Fig. 15). The high strain hardening rate achieved in the AlCuSc-C alloy is supposed to be due to the strong hindering of moving dislocations by the high-density nanosized atom complexes enriched with Cu, Sc, and excess vacancies, which enhances the accumulation of dislocations. When the moving dislocations encounter complexes, an extra force is needed to break complexes, resulting in a pinning effect on the moving dislocations.

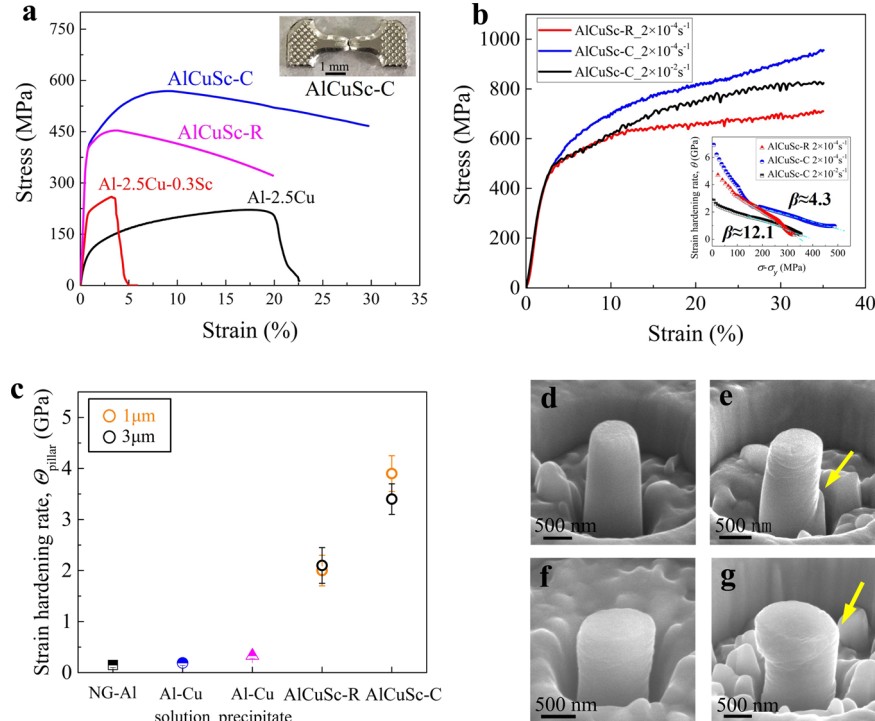

**Fig. 5 Mechanical properties and deformation behavior of AlCuSc-R and AlCuSc-C alloys during tensile testing and micropillar compression test.**
**a** Tensile engineering stress-strain curves of the AlCuSc-R and AlCuSc-C alloys, in comparison with peak-aged coarse-grained Al-Cu and Al-Cu-Sc alloys with plate-like $\theta'$ precipitates[55]. Inset is the fractured tensile specimen of the AlCuSc-C alloy. **b** Compressive engineering stress-strain curves of the 1-μm-diameter micropillars of the AlCuSc-R and AlCuSc-C alloys at different compressing strain rates. Inset is the corresponding Kocks-Mecking plots, $\beta$ is characteristic of dynamic recovery and defined as the slope marked by the dash-doted line. **c** Calculated strain hardening rate of the AlCuSc-R and AlCuSc-C micropillars, in comparison with that of the NG Al micropillar, solid solution-treated Al-Cu micropillar, and peak-aged coarse-grained Al-Cu micropillar. The error bars represent standard deviations from the mean for sets of three tests. **d** SEM image of the as-fabricated 1-μm-diameter AlCuSc-C micropillar. **e**, **f** are SEM images of the AlCuSc-R and AlCuSc-C micropillars after compression at a strain rate of $2 \times 10^{-4}$ s$^{-1}$, respectively. **g** SEM image of the AlCuSc-C micropillar after compression at a strain rate of $2 \times 10^{-2}$ s$^{-1}$.

This process would increase the opportunities for dislocations to interact with each other, enhancing the accumulation of dislocations in the grain interior and thus the strain hardening ability[57].

The difference in plastic deformation behavior between AlCuSc-C and AlCuSc-R alloys can also be observed in the compressed micropillar. Figure 5d representatively shows a well-machined micropillar before compression. After compression, a bunch of slip bands are visible in the micropillar of the AlCuSc-R alloy (Fig. 5e), indicative of an inhomogeneous deformation and local strain concentration. In contrast, the compressed AlCuSc-C alloy micropillar shows a uniform deformation (Fig. 5f).

More interestingly, the AlCuSc-C alloy shows a reduction in flow stress when a higher strain rate ($2 \times 10^{-2}$ s$^{-1}$) was applied (see Fig. 5b), implying a negative strain rate sensitivity (SRS). The negative SRS was further verified by nanoindentation tests (see Supplementary Fig. 16a). In contrast, the other three NG alloys in the present work all displayed a positive SRS. This discrepancy highlights a distinct time-sensitive pinning effect on dislocations exerted by the atom complexes enriched with Cu, Sc, and excess vacancies. Negative SRS is usually observed in some coarse-grained Al-Mg-based alloys containing a high Mg content and is widely rationalized by the dynamic solute (cluster)-dislocation coupling, or termed dynamic strain aging[58]. In the present AlCuSc-C alloy, the underlying mechanism for negative SRS can be fundamentally akin to the dynamic strain aging, where the (Cu, Sc, vacancy)-rich atomic complexes serve as obstacles for dislocation slip. During deformation, the (Cu, Sc, vacancy)-rich atomic complexes can be decomposed by dislocation shearing. Due to the strong binding

tendency among Cu atom, Sc atom, and excess vacancies, new (Cu, Sc, vacancy)-rich atomic complexes can be regenerated, pinning the slip of dislocations. However, when a strain rate as high as $2 \times 10^{-2}$ s$^{-1}$ was used, there may be not enough time for new complexes to be regenerated before subjecting to new dislocation slip. Thus, the pinning effect of (Cu, Sc, vacancy)-rich atomic complexes on dislocations is reduced and the strain hardening ability of the material is weakened. The homogeneous deformation is therefore edged out by a shear-like localized deformation, see the evolving deformation morphology from Fig. 5f to g. However, it should be mentioned that no significant serration could be observed in the stress-strain curve of the present AlCuSc-C alloy, which is different from the typical Portevin-Le Chartelie (PLC) effect. It means that a further study on how the (Cu, Sc, vacancy)-rich atomic complexes influence the deformation behavior is necessary for the future.

Besides the 1-μm-diameter micropillars, 3-μm-diameter micropillars were also tested. The aim was to exclude the possibility that a sample size effect incurred the different plastic deformation behaviors between the AlCuSc-C and AlCuSc-R alloy micropillars[56]. The additional results (Supplementary Fig. 16b) basically had the same implications as the above results. Thus, it can be concluded that the stronger strain hardening capability achieved in the AlCuSc-C alloy should be related to the presence of atomic complexes.

## Discussion
In this work, we show an effective strategy to stabilize supersaturated solid solutions in the NG Al-Cu alloys and promote

their thermal stability to an applicable level. The suppression of low-temperature unfavorable precipitation is effectuated by introducing high-density vacancies and simultaneously Sc microalloying atoms. Thermodynamically stable (Cu, Sc, vacancy)-rich atomic complexes are self-organized that stop the Cu diffusion effectively and trap the solute atoms within the Al matrix. In this mechanism, the vacancies with a substantially higher concentration are necessary to ensure that excess vacancy can be involved in most atom complexes, which significantly enhances the thermal stability of atom complexes. HPT at liquid nitrogen temperature is crucial to produce such high-density vacancies. Microalloying element Sc that has a strong binding tendency with Cu is also required in triggering the formation of (Cu, Sc, vacancy)-rich atomic complexes and suppressing the annihilation of excessive vacancies. We demonstrate that the (Cu, Sc, vacancy)-rich atomic complexes enable the NG Al-Cu alloy with not only a satisfactory resistance to Cu precipitation but also a stronger strain hardening capability. This vacancy-mediated complexing strategy concurrently settles two bottleneck problems inherent to NG Al alloys, i.e., uncontrollable low-temperature precipitation of stable precipitates and insufficient plastic deformability. The design concept of high-density vacancies together with microalloying may be applied to other NG metallic alloys.

The highly-exalted strain hardening capability derived from the (Cu, Sc, vacancy)-rich atomic complexes is of high interest for the community of NG metallic materials. A big challenge for the application of NG metallic materials is the poor ductility[16,17]. Some tactics have been proposed to improve the ductility of bulk NG metals, including GB chemical/structural optimization and multi-length heterogeneity design[24,26]. In particular, dispersing nanosized particles in the grain interior via precipitation is an effective approach, because the intragranular particles can generate, pin down and thus accumulate dislocations within the grains[18,26]. But uncontrollable intergranular precipitation preferentially occurred in the NG metals at low temperature (even room temperature)[20,22], suppressing the intragranular precipitation. Here in this work, we propose a vigorous approach, i.e., solute-vacancy complexes, to raise the strain hardening capability of the NG AlCuSc-C alloy to a high level. These solute-vacancy complexes are somewhat akin to solute clusters, with slightly smaller sizes and similar distribution. The Al alloys microstructurally featured with solute clusters have been found to possess a strain hardening capability greater than their counterparts reinforced by precipitates[59,60]. In the present NG AlCuSc-C alloy, the interactions between larger atom complexes and mobile dislocations are rather complex that continuous vacancy/solute dissociation and vacancy/solute rebinding may happen.

There remain several unanswered questions concerning the (Cu, Sc, vacancy)-rich atomic complexes. Firstly, the self-organizing process of such complexes is still unclear. In the future, molecular dynamic simulation can be adopted to simulate the formation process of (Cu, Sc, vacancy)-rich atomic complexes, the further growth of Cu-Sc-double vacancies into larger complexes, and the diffusion of Cu and Sc solutes under the influence of high concentration of vacancies. Relevant theoretical frameworks, including thermodynamics and kinetics for the complex formation, evolution, and dissociation, should be developed for the complex design and stability optimization. These questions may inspire a series of research focusing on the fundamentals of the solute-vacancy complex in the future. Our approach that exploits great strain hardening capability combined with high thermal stability, is one that we believe can be widely applied to NG metallic alloys prepared by SPD and probably also the alloys in far-from-equilibrium conditions.

## Methods

**Fabrication of the materials**. Alloys with compositions of Al-2.5 wt.%Cu and Al-2.5 wt.%Cu-0.3 wt.%Sc were respectively melted and cast in-stream argon, by using 99.99 wt.% pure Al, 99.99 wt.% pure Cu, and master Al-2.0 wt.% Sc alloy. The cast alloys were homogenized at 450 °C for 4 h and the solution was treated at 590 °C for 3 h. Disks with 10 mm in diameter and1.2 ± 0.1 mm in thickness were cut from the ingot for HPT processing. The HPT experiments were performed at room temperature and in liquid nitrogen, respectively, under a pressure of 6 GPa and with a rotational speed of 1 rpm for ten revolutions. During HPT processing in liquid nitrogen, the sample and anvils were immersed in the cooling medium (liquid nitrogen, ~77 K), which enabled the suppression of temperature rise during high shear strain deformation. After HPT processing, the samples were immediately artificially aged at 125 °C for a series of times in an oil bath.

**Microstructure characterization**. To study the microstructures at the nanoscale, transmission electron microscopy (TEM) and high-angle annular dark-field (HAADF) scanning transmission electron microscopy (STEM) were carried out by using JEOL 2100 F operating at 200 kV and Cs-corrected FEI G2-300 Titan operating at 300 kV. TEM orientation mapping was performed using the JEOL 2100 F at 200 kV, equipped with the NanoMagas ASTAR system. TEM foils were prepared by following standard electro-polishing techniques for Al alloys. The X-ray diffraction patterns were measured at the beamline 14B1 of the Shanghai Synchrotron Radiation Facility (SSRF). The X-ray wavelength was 0.68879 A˚ with an energy of 18 KeV, and the spot diameter is 0.4 mm. The diffraction patterns were recorded by using the 9910 point detector, with a step size of 0.01° and a dwell time of 0.5 s. The standard sample $LaB_6$ was tested to evaluate the instrumental broadening. The Convolutional Multiple Whole Profile (CMWP) procedure, developed by ref. [61], was utilized to calculate the dislocation density from the measured XRD patterns. Atom probe tomography (APT) experiments were performed using a Local Electrode Atom Probe (LEAP 5000XS) from CAMECA. Needle-shaped specimens were prepared by a standard lift-out method using a Helios NanoLab Dual-Beam focused ion beam (FIB) from FEI. The end radius of curvature of the needles were less than 50 nm. The APT analyses were performed in laser mode, at a set-point temperature of 30 K and under ultrahigh vacuum at a pressure below $2.0 \times 10^{-9}$ Pa ($1.5 \times 10^{-11}$ Torr). The laser kept a pulse repetition rate of 250 kHz and the energy was calibrated for each individual tip to yield an equivalent pulse fraction in voltage mode of 20%, which corresponded to laser energy between 75 and 160 pJ. The detection rate was set to have 0.5% of the applied laser pulses resulting in an evaporation event. Reconstruction and analysis of the acquired data were carried out using IVAS™ 6 software. The solute clusters were defined following the cluster-finding procedure in refs. [53,60], where the maximum distance between neighboring solute atoms in a cluster, $d_{max}$, and the minimum number of atoms in a cluster, $N_{min}$, are the key user-defined parameters. The $d_{max}$ was determined as the distance at which the difference of cumulative probabilities of the first-order nearest neighbor distance (1NND) between the experimental and random data is greatest. The parameter $N_{min}$ was determined by comparing the size distribution of the random solute clusters with that of the experimentally detected ones. The partial radial distribution function (RDF) technique was applied to LEAP tomographic data, providing a measure of solute-solute clustering[48]. A partial RDF at a radial distance $r$ is defined as the average concentration distribution of component $i$ around a given solute species X, $\langle c_i^X(r) \rangle$, normalized to the overall concentration of $i$ atoms, $c_i^0$, in the sampled volume:

$$\text{Partial RDF} = \frac{\langle c_i^X(r) \rangle}{c_i^0} = \frac{1}{c_i^0} \sum_{k=1}^{N_x} \frac{N_i^k(r)}{N_{tot}^k(r)} \qquad (1)$$

where $N_i^k(r)$ is the number of $i$ atoms in a radial shell around the kth X atom that is at the center of a shell with radius $r$, $N_{tot}^k(r)$ is the total number of atoms in this shell, $N_X$ is the number of X atoms in the analyzed volume. Partial RDF values of unity indicate perfect random distribution, and partial RDF values larger than unity describe clustering of the species $i$ and X.

Positron annihilation lifetime spectroscopy (PALS) experiments were performed to measure the type and concentration of defects in materials[31,33]. The spectra comprise at least 2 million counts and the time resolution of the system is about 208 ps. A program MELT 4.0 was used to analyze the positron lifetime[31]. Coincidence Doppler broadening (CDB) measurements of the positron annihilation radiation were performed using two high-purity Ge detectors[51]. The energies of annihilating γ-ray pairs (denoted by $E_1$ and $E_2$) were simultaneously recorded by the two detectors located at an angle of 180° relative to each other. The difference in energies of the two γ-rays $\Delta E = E_1 - E_2$ is expressed as $cP_L$ and the total energy $E_t = E_1 + E_2$ is expressed as $2m_0c^2 - E_B$ (neglecting the thermal energies and chemical potentials), where $P_L$ is the longitudinal component of the positron-electron momentum along the direction of the γ-ray emission, c is the speed of light, $m_0$ is the electron rest mass, and $E_B$ is the electron binding energy. A total count of more than $2 \times 10^7$ for each measurement was accumulated for 12 h.

**Mechanical characterization**. Vickers hardness measurements were performed on a Buehler Wilson Hardness Tester (VH3100) using an applied load of 5 kg for 15 s on samples polished to at least a 1 μm surface finish. Specifically, each hardness value was determined by taking the average of three separate measurements recorded at the edge of the HPT specimen. Tensile tests were conducted at room temperature at a strain rate of $10^{-4} \text{ s}^{-1}$, using a computer-controlled testing

machine operating with a constant displacement of the specimen grips. A laser extensometer, P-50 by Fiedler Optoelectronics, was used to accurately measure the strain. Tensile specimens had 1 mm width and 0.8 mm thickness with a 2-mm-gauge length. Tensile specimens were taken from the area at a half-radius distance from the center of processed disks. At least three samples were tested for each alloy. Nanoindentation tests were performed using a TI950 TriboIndenter (Hysitron, Minneapolis, MN) with a standard Berkovich tip at room temperature, following the Oliver-Pharr method[62]. Micropillars were fabricated by a focused ion beam facility (FIB) with 30 kV/1.5 pA as the final milling condition. The aspect ratio (height/diameter) of the pillar was kept between 2.5:1 and 3.5:1. We conducted SEM in situ compression tests at room temperature using a PI 87 PicoIndenter (Hysitron Inc.) with diamond punch/gripper inside an FEI Quanta 600 FEG scanning electron microscope, under displacement-control mode and at a strain rate of $2 \times 10^{-4}$–$2 \times 10^{-2} \, \mathrm{s}^{-1}$.

**Differential scanning calorimetry (DSC)**. Thermal analysis was performed using a differential scanning calorimeter (DSC, Mettler Toledo Tga/DSC3+) at a heating rate of 10 °C/min in a flowing $\mathrm{Ar_2}$ atmosphere from ambient temperature to 400 °C. Three specimens were measured for each thermogram, with each specimen of approximately 10 mg. The samples were put in Al crucibles under tight-fitted inverted lids, with an empty Al crucible as the reference. The DSC thermograms were corrected by subtracting a baseline run with an empty Al crucible.

**Density functional theory (DFT) calculations**. The binding energies between solute atoms (Cu and Sc) and a vacancy in the Al matrix were calculated based on the DFT method[49]. According to the physical definition, binding energy equals in quantity to the work used to separate the solute atoms and vacancies by an "infinite" distance. Considering an atomic Al supercell containing $N$ atom sites, if the numbers of Cu solute atoms, Sc solute atoms, and vacancies are $x$, $y$, and $z$, respectively, the binding energy $E_b$ is expressed as:

$$-E_b = E\left(\mathrm{Al}_{N-x-y-z}\mathrm{Cu}_x\mathrm{Sc}_y V_z\right) + (x+y+z-1)E(\mathrm{Al}_N) - xE(\mathrm{Al}_{N-1}\mathrm{Cu})$$
$$-yE(\mathrm{Al}_{N-1}\mathrm{Sc}) - zE(\mathrm{Al}_{N-1}V) \tag{2}$$

where the minus is used to keep the binding energy consistent with the convention in literature, i.e., positive binding energy indicates a favorable binding[49]. Symbol $V$ represents vacancy in a super cell, $E$ is the ground state energy of a super cell with atom composition given by the subscripts in parenthesis, e.g., $E\left(\mathrm{Al}_{N-x-y-z}\mathrm{Cu}_x\mathrm{Sc}_y V_z\right)$ denotes the energy of a supercell containing $N$-$x$-$y$-$z$ Al atoms, $x$ Cu atom, $y$ Sc atoms and $z$ vacancies. The DFT calculations were conducted by the Vienna ab-initio Simulation Package (VASP) with a plane-wave-based pseudopotential method. The super cell was constructed as a $4 \times 4 \times 4$ fcc lattice (i.e., 256-atom supercell) with three-dimensional periodic boundary conditions. The projector augmented wave (PAW) potential[63] was used to describe the Coulomb interaction of ion cores with the valence electrons, and the local density approximation (LDA) was used for the electronic exchange correlation[64]. The maximum K-point grid of the Brillouin zone integration was an $8 \times 8 \times 8$ automatic Monkhorst-Pack grid, and the energy cut-off was set as 400 eV. Full energy relaxation of the super cell was performed with respect to cell volume, shape, and ion positions. By substituting into Eq. (2) the final ground state energies of the relaxed supercells, the binding energy between solute atoms and vacancies could be precisely determined. To characterize the lattice distortion of the super cell after relaxation, a structural order parameter $\Theta$[65] of the super cell was calculated, of which a larger value indicated a greater lattice distortion. The distribution of the charge density around the solute atoms and vacancies of the relaxed structures were plotted by the software VESTA.

## Data availability
The datasets generated during and/or analyzed during the current study are available from the corresponding author on request.

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

## Acknowledgements

This work was supported by the National Natural Science Foundation of China (Grant nos. 51625103, 51790482, 51722104, 51761135031, and 52001249) and the 111 Project of China (BP2018008). This work is also supported by the International Joint Laboratory for Micro/Nano Manufacturing and Measurement Technologies. Y.P. acknowledges the support from the National Natural Science Foundation of China (Grant nos. 51771085, 51571104, and 51801087). The authors would like to acknowledge the support of the National infrastructure of Norway, MiMaC (Project number: 269842). The authors would thank Mr. Pål J. Skaret for his assistance during HPT experiments and tensile testing. The authors would also thank Y. Z. Chen at Northwestern Polytechnical University and Ruben Bjørge at SINTEF Materials and Chemistry for ASTAR-TEM experiments. The authors are grateful to Prof. S.W. Guo and Dr. J. Li at XJTU for their great assistance with TEM analysis.

## Author contributions

G.L., Y.L., H.J.R., and J.S. initiated and supervised the project. S.W. prepared the alloys and carried out most of the microscopy. S.W. and C.Y. performed the property measurements. H.S.S., Y.L., and C.L. did the APT examination and data analysis. P.Z. and P.C. performed the synchrotron XRD experiments. S.W. and J.Z. carried out the positron annihilation measurements. B.C. and Y.P. performed the HAADF examinations. J.B. did the DFT calculations. All authors extensively discussed the data. G.L., Y.L., and S.W. wrote the paper.

## Competing interests

The authors declare no competing interests.
