## [Peer Review File · Nature Communications]

Title: Freezing solute atoms in nanograined aluminum alloys via high-density vacanciesREVIEWER COMMENTS

Reviewer #1 (Remarks to the Author):

This is an important contribution, definitely deserving publication.

The authors have investigated and interrogated the unexpected temperature response of select Al alloys with high vacancy concentrations, thereby opening a new path for alloy development.

I have no hesitation in recommending acceptance. I would however recommend that the manuscript be given a thorough editorial review to ensure that the use of English is consistent.

Reviewer #2 (Remarks to the Author):

The article reports on nanograined aluminum prepared by severe plastic deformation. The authors detected a counterintuitive strategy to stabilize supersaturated solid solution in nanograined Al-Cu-Sc alloys resulting in some additional strain hardening potential. This result is definitely interesting and worth to report. The used appropriate experimental methods such as atom probe tomography and aberration-corrected scanning transmission electron microscopy, mechanical testing and DFT simulations are state of the art and appropriate. However, they explain their finding via ultrahigh-density vacancies in combination with Sc microalloying. The proposed concept is not unrealistic, but the storyline raises some major questions which weakens the proposed concept in such extent that I am not convinced, yet. Nevertheless, there are dedicated and major issues upon this line which may be answered by the authors in a mandatory revision:

Page 5: It is not clear why the vacancies should collapse to such large clusters upon PIPS preparation. According to the authors the Sc-V clusters would even resist temperatures > 200 °C. There is need to provide proof on that, otherwise also the V clusters could be annihilation sites for PA and vacancies as well, keeping the system super-fast at equilibrium V fraction. The origin and the role of these large V-Clusters (pores) need to be much better explained and potentially also considered in the discussion.

Page 7: I can see that precipitation is suppressed in II-C at 125 °C. However, I do not think that only excess vacancies play a role at this T. In the NG alloy (and probably in a CG alloy) equilibrium diffusion and GB diffusion could be sufficient to form precipitates at the GB when aged for 6h. However, the question remains why II-C shows this interesting behavior.

Page 13: The authors should show measurements (APT, TEM) that support the statement, "Therefore, there are not enough vacancies to form a high fraction of (Cu, Sc, vacancy)-rich atomic complexes containing a large fraction of vacancies in the II-R alloy."

Page 13: The statement "As a consequence, the Sc effect in slowing down Cu diffusion is overwhelmed

by the high-density dislocations and the catastrophic θ -precipitation can still be happening during room temperature storage.”, is not clear to me. In II-C the V are trapped and in II-R they annihilate at dislocations. Why should there be a difference in the θ formation at RT storage? This would be both guided by the equilibrium V fraction. Does the θ formation for II-R really happens during RT storage? This need experimental proof.

In the following minor suggestions to improve the draft and questions which may further support the case are given:

The statement “catastrophic” precipitation might be relevant for Al-Cu alloys, but heterogenous precipitation is not always negative in Al alloys and I would not overpower the term catastrophic here.

The notations II-C etc. are not intuitive and can also not be found at a dedicated place in the paper. One always needs to look this up, which distracts from the flow during reading the paper.

Page 6: It should be mentioned what role the PA lifetime in Sc-clusters play? Was CDB done to verify if Sc is located next to the vacancies?

Page 6: Why is the lifetime in NC Al-Cu generally that high? What positron sinks are active there?

Page 9-10: I cannot see the formation of an enrichment up to 5% Cu in GP-zones after aging from Fig 4.

Page 11: Fig 4c looks like typical L12 Al₃Sc clusters. It is not clear to me why the FFT is not interpreted like that.

Page 15: The representation of strain hardening rate in Fig. 5c is misleading. There is no physical valid curve through the different values. Why do the authors not use KM-plots derivate from Fig 5a to calculate strain hardening rates?

Page 17: The statement “It is soundly proved that the stronger strain-hardening capability in the II-C alloy should be related to atomic complexes.”, is a bit too strong from my opinion. This is only discussed based on indirect conclusions.

SFig.8: The only difference visible to me in the NN is statistics. There is no clear difference.

Reviewer #3 (Remarks to the Author):

The authors present a highly original and extensive study where they report and explain the improved

thermal stability of a nanostructured Al-Cu-Sc alloy in the presence of a supersaturation of vacancies. Explanations for improved thermal stability include experimental measurements (Positron Annihilation Spectroscopy) and theoretical calculations (density functional theory calculations of vacancy pairs with Cu and Sc atoms). Connection between the solute-vacancy complexes and the mechanical properties (e.g. strain hardening) is also well-made. The work is likely to be impactful.

The present reviewer suggests a few changes and clarifications:

- The authors need to reconsider the many superlatives they have used throughout the manuscript. How are “ultrahigh vacancy densities” different from high vacancy densities? The use of “ultrahigh thermal stability” on page 7 is especially objectionable for an alloy microstructure that is more thermally stable than other nanostructured alloys but not very thermally stable in an absolute sense since many Al alloys will have significantly higher thermal stability than the alloys under consideration.
- Page 2 – line 56 – replace low dimensional materials with low dimension material geometries since the authors are referring to APT needles and TEM foils instead of low dimensional materials such as graphene and nanoribbons.
- Figure 1 caption – replace “gains” with “grains”

Amendments and Corrections According to the Editor/Referee's Comments

Thanks for the editor/reviewers' valuable comments and detailed suggestions on our manuscript, which are of great help for the further improvements in quality of our work. According to the reviewers' suggestions/comments, we have made a thorough revision of the manuscript. All the changes have been highlighted by yellow color in the revised manuscript. In the following response, we have replied the comments by the reviewers and explained the corresponding revisions.

REVIEWER COMMENTS

Reviewer #1 (Remarks to the Author):

This is an important contribution, definitely deserving publication.

The authors have investigated and interrogated the unexpected temperature response of select Al alloys with high vacancy concentrations, thereby opening a new path for alloy development. I have no hesitation in recommending acceptance. I would however recommend that the manuscript be given a thorough editorial review to ensure that the use of English is consistent.

Response: Thanks for the reviewer's positive judgement. We have done a thorough reading and language cleaning of the manuscript to ensure that the use of English is consistent. For examples, "nano-grained" has been replaced by "nanograined", and "grain boundaries" has been consistently abbreviated as GBs. Other inconsistencies have also been corrected in the same way.

Reviewer #2 (Remarks to the Author):

The article reports on nanograined aluminum prepared by severe plastic deformation. The authors detected a counterintuitive strategy to stabilize supersaturated solid solution in nanograined Al-Cu-Sc alloys resulting in some additional strain hardening potential. This result is definitely interesting and worth to report. The used appropriate experimental methods

such as atom probe tomography and aberration-corrected scanning transmission electron microscopy, mechanical testing and DFT simulations are state of the art and appropriate. However, they explain their finding via ultrahigh-density vacancies in combination with Sc microalloying. The proposed concept is not unrealistic, but the storyline raises some major questions which weakens the proposed concept in such extend that I am not convinced, yet. Nevertheless, there are dedicated and major issues upon this line which may be answered by the authors in a mandatory revision:

Question 1. Page 5: (1) It is not clear why the vacancies should collapse to such large clusters upon PIPS preparation. According to the authors the Sc-V clusters would even resist temperatures > 200 °C. (2) There is need to provide proof on that, otherwise also the V clusters could be annihilation sites for PA and vacancies as well, keeping the system super-fast at equilibrium V fraction. The origin and the role of these large V-Clusters (pores) need to be much better explained and potentially also considered in the discussion.

Response: (1) Thanks for the reviewer's valuable comment and helpful suggestions. Actually, the formation of large V-clusters in the form of voids upon PIPS preparation while the Sc-V clusters resisting temperatures > 200 °C could be rationalized as follows. During ion milling, Ar ions were bombarded at a glancing angle of 4° . At an energy of 3 keV, the penetrating depth of Ar ions in the Al is about 3 nm, as evaluated using SRIM¹. This may limit the heavily-damaged region within the surface part of the TEM thin-foil. However, surface ion bombardment can usually cause composition fluctuations beyond the penetrating depth, which is well known as "long-range effects"². Such fluctuations are commonly associated with two mechanisms: the enhanced atomic migration due to beam heating (up to $100^\circ\text{C} - 200^\circ\text{C}$)³ and the enhanced atomic migration as a result of ballistic collisions and energy dissipation of collision cascades. In present AlCuSc-C (*i.e.*, termed II-C in the original version, see **Question 6**) thin-foil, a high density of (Cu, Sc, vacancy)-rich atomic complexes exist in the matrix. Under Ar ions irradiation, the ballistic collision between Ar ions and Cu (and/or Sc) atoms can break the bonds between solute atoms and vacancies in the atomic complexes, leaving a high

quantity of free vacancies retained in the matrix. The abundant free vacancies, driven by the accompanying high temperature, induce the aggregation of vacancies, leading to a high density of voids (or large vacancy clusters in present work). Similar phenomenon is commonly observed in ion-irradiated metallic materials at high temperatures⁴, where a great number of vacancies are triggered by the ion-irradiation. In contrast, no voids could be observed in the TEM foils prepared by electro polishing method, which confirms that the formation of voids observed in Fig. 1c was due to the irradiation of ions during ion milling. It should be noted that, in the other three alloys in the present work, no voids could be observed in TEM samples prepared by ion milling, which should be attributed to the much lower density of vacancies. In the revised manuscript (please see page 5), we have added the following sentence: “Under Ar ion bombardment, the collision cascades and induced temperature increase cause the aggregation of vacancies into voids”.

(2) Based on above discussions, it can be concluded that the voids observed in Fig. 1c are generated during the ion milling process, which will not influence the PA analysis. During the positron annihilation lifetime spectroscopy (PALS) test, positrons from the decay of ²²Na are injected into the specimen, which will reach thermal equilibrium in the sample and eventually annihilate with an electron, producing γ -quanta⁵. The lifetime of a positron trapped in a vacancy-type defect is enhanced relative to the bulk, owing to the reduced electron density at the defect site^{6,7}. This indicates that injected positrons do not change the characteristic and distribution of vacancy-type defects. Thus, injecting positrons into the bulk specimen will not lead to the voids (large vacancy clusters) as observed in the AlCuSc-C alloy upon PIPS preparation. Vacancies in the (Cu, Sc, vacancy)-rich atomic complexes can act as annihilation sites for positrons. Similar behavior has been reported in Al-Cu-Mg alloy, where vacancies in Mg-vacancy and Cu-Mg-vacancy complexes can act as annihilation sites for positron annihilation⁸.

Question 2. Page 7: I can see that precipitation is suppressed in II-C at 125 °C. However, I do not think that only excess vacancies play a role at this T. In the NG alloy (and probably in a CG alloy) equilibrium diffusion and GB diffusion could be sufficient to form precipitates at

the GB when aged for 6h. However, the question remains why II-C shows this interesting behavior.

Response: Thanks for the reviewer's professional question. We agree that only excess vacancies could not suppress the precipitation of Al-Cu precipitates in the AlCuSc-C (or II-C) alloy during ageing at 125°C. Instead, the suppression is due to the formation of (Cu, Sc, vacancy)-rich atomic complexes, which is a result of a combined effect of excess vacancies and Sc microalloying atoms in the alloy. That is also what we are trying to demonstrate in the manuscript. For those alloys processed by room-temperature HPT (AlCu-R or I-R and AlCuSc-R or II-R, see Question 6), and Sc-free alloys processed by cryogenic HPT (AlCu-C or I-C, see Question 6), equilibrium diffusion and GB diffusion are indeed sufficient to form precipitates at the GB, which is evidenced by the formation of stable θ -Al₂Cu precipitates at GBs and in the bulk of grains (Fig. 2 and Supplementary Fig. 5). However, in the AlCuSc-C alloy, the formation of high-density, low-energy (Cu, Sc, vacancy)-rich atomic complexes can significantly reduce the amount of available free Cu solute atoms in the bulk that could diffuse to GBs, forming GB precipitates. On the other hand, the diffusivity of Sc atoms is several orders of magnitude slower than the Cu atoms and the vacancies⁹⁻¹¹, which will significantly reduce the diffusivity of the atomic complexes and therefore the mobility of Cu atoms bonded in such complexes. The APT results (see Fig. 4 in the revised manuscript) provide sound evidence. After natural aging for 6 months, the average Cu concentration within the grain is about 1.05 at.% (or ~ 2.5 wt.%) in the AlCuSc-C alloy, almost the same as the nominal Cu addition, indicative of a nearly full solid solution of Cu atoms. Although a slight Cu and Sc segregation at the GBs could be observed, the thickness of solute depletion zone at either side of GBs is rather thin (~ 5 nm). This indicates that the diffusion of Cu and Sc atoms towards GBs is restricted to a very thin region adjacent to the GB. After aging at 125°C for 6 h, the Cu concentration within the grain interior still keeps at the level of ~ 0.86 at.%, showing that the (Cu, Sc, vacancy)-rich atomic complexes are rather stable.

Question 3. Page 13: The authors should show measurements (APT, TEM) that support the statement, “Therefore, there are not enough vacancies to form a high fraction of (Cu, Sc, vacancy)-rich atomic complexes containing a large fraction of vacancies in the II-R alloy.”

Response: Thanks for the reviewer’s helpful suggestion. Additional APT and TEM images are provided in the revised manuscript. We have used APT to characterize the Al-Cu-Sc alloy after room-temperature HPT (AlCuSc-R or II-R). A representative APT reconstruction of the as-processed AlCuSc-R alloy is shown in **A-Figure 1a**. No atom clusters or precipitates could be detected in the reconstructed APT map. However, the corresponding RDF analysis in **A-Figure 1b** shows a slight enrichment of Cu atoms around Sc atoms, implying that (Cu, Sc, vacancy)-rich atomic complexes have also formed during room-temperature HPT. A comparison to the RDF map in Supplementary Fig. 9 shows that the normalized concentration of Cu around Sc atoms is much lower than that of the AlCuSc-C (or II-C) alloy, indicating that the number density of (Cu, Sc, vacancy)-rich atomic complexes in the AlCuSc-R alloy is much lower.

Representative HAADF-STEM images of the AlCuSc-R alloy stored at room temperature for 6 months are presented in **A-Figure 2**. As shown in **A-Figure 2a**, a large number of coarse θ -Al₂Cu precipitates have formed. No trace of (Cu, Sc, vacancy)-rich atomic complexes could be observed in the high-resolution HAADF-STEM image (**A-Figure 2b**). Revision is made accordingly, please see Supplementary Fig. 14.

A-Figure 1 APT analysis of the as-processed AlCuSc-R alloy. **a** Representative APT reconstruction of the as-processed AlCuSc-R alloy, where Cu and Sc atoms are labelled with brown and blue colors, respectively. **b** The RDF curves of Cu and Sc atoms around Sc atoms in the as-processed AlCuSc-R alloy.

A-Figure 2 HAADF-STEM images of the AlCuSc-R alloy stored at room temperature for 6 months. **a** Typical HAADF-STEM image of the AlCuSc-R alloy stored at room temperature for 6 months showing apparent θ precipitation. **b** High-resolution HAADF-STEM image viewed along $\langle 100 \rangle_{\text{Al}}$ showing no apparent (Cu, Sc, vacancy)-rich atomic complexes.

Question 4. Page 13: The statement “As a consequence, the Sc effect in slowing down Cu diffusion is overwhelmed by the high-density dislocations and the catastrophic θ -precipitation can still be happening during room temperature storage.”, is not clear to me. In II-C the V are trapped and in II-R they annihilate at dislocations. 1. Why should there be a difference in the

θ formation at RT storage? This would be both guided by the equilibrium V fraction. 2. Does the θ formation for II-R really happens during RT storage? This needs experimental proof.

Response: Thanks for the reviewer's valuable questions. (1) Probably, we have not explained clearly enough in the manuscript. As quantitatively given in Supplementary Fig. 3, in both AlCuSc-R (or II-R) and AlCuSc-C (or II-C) alloys, high-density dislocations, in the range of $4.0 \times 10^{14} \text{ m}^{-2}$ - $7.1 \times 10^{14} \text{ m}^{-2}$, exist. Dislocation cores can act as fast diffusing channel for solute atoms since the disorders can effectively lower the activation energy for diffusion^{12,13}, leading to enhanced effective diffusion coefficient. Dislocations can also act as heterogeneous nucleation sites for precipitates^{11,14}. Both contribute to the significantly-accelerated precipitation kinetics. In the AlCuSc-C alloy, during cryogenic HPT process, a much higher concentration of vacancies could be generated, which enhanced the formation of low-energy (Cu, Sc, vacancy)-rich atomic complexes. The formation of such high-density atomic complexes significantly stabilizes Cu atoms and vacancies from diffusing to dislocations and forming precipitates. In contrast, much lower concentration of vacancies could be generated in the AlCuSc-R alloy during room-temperature HPT. As a result, only a limited number of (Cu, Sc, vacancy)-rich atomic complexes could form (**Question 3, A-Figure 1 and A-Figure 2**). Under this condition, there is a large fraction of single Cu atoms and vacancies in the matrix. The high-density dislocations can markedly enhance the precipitation of supersaturated Cu, leading to unstable θ -precipitation during room-temperature storage. Revision is made accordingly, please see page 14 in the revised manuscript.

(2) The experimental proof of θ formation for the AlCuSc-R alloy during room-temperature storage is provided in Supplementary Fig. 6.

In the following minor suggestions to improve the draft and questions which may further support the case are given:

Question 5. The statement “catastrophic” precipitation might be relevant for Al-Cu alloys, but heterogeneous precipitation is not always negative in Al alloys and I would not overpower the term catastrophic here.

Response: Thanks for the reviewer’s professional and valuable suggestion. We agree with the reviewer that heterogeneous precipitation is not always negative in SPD-processed Al alloys, and the word “catastrophic” should be used more discreetly. In the revised manuscript, the application of word “catastrophic” has been limited to the precipitation of coarse stable θ -Al₂Cu precipitates at GBs.

Question 6. The notations II-C etc. are not intuitive and can also not be found at a dedicated place in the paper. One always needs to look this up, which distracts from the flow during reading the paper.

Response: Thanks for the reviewer’s valuable suggestion. In the revised manuscript, we have made corresponding revisions: the as-processed Al-2.5 wt.%Cu-0.3 wt.%Sc alloy by HPT at liquid nitrogen/cryogenic temperature is now denoted as **AlCuSc-C** instead of II-C. The Sc-free Al-2.5 wt.%Cu alloy by HPT at cryogenic temperature is denoted as **AlCu-C** instead of I-C. The Al-2.5 wt.%Cu alloys with and without 0.3 wt.%Sc addition by traditional room-temperature (298 K) HPT are denoted as **AlCuSc-R** and **AlCu-R**, respectively, rather than II-R and I-R.

Question 7. Page 6: It should be mentioned what role the PA lifetime in Sc-clusters play? Was CDB done to verify if Sc is located next to the vacancies?

Response: Thanks for the reviewer’s valuable suggestions. In order to reveal the role of the PA lifetime in (Cu, Sc, vacancy)-rich atomic complexes (*i.e.*, Sc-clusters) and the Sc locations in the atomic complexes with regards to vacancy, we have additionally done Coincidence Doppler broadening (CDB) analyses of the AlCuSc-C (or II-C) alloy, following the analysis

as in Refs.^{8,15}. CDB experiments for pure Al (well-annealed), pure Cu, pure Sc and the AlCuSc-C alloy were performed, respectively. The CDB spectra are presented in **A-Figure 3a** for comparison, and the corresponding CDB ratio curves normalized to pure Al are presented in **A-Figure 3b**. The CDB ratio curve of the AlCuSc-C alloy shows a mixture of the characteristic Cu signal and Sc signal: not only a long tail in the high momentum region ($> 15 \times 10^{-3} m_0c$) due to Cu electrons but also a hump around $10 \times 10^{-3} m_0c$ due to Sc electrons. The agreements indicate that a large fraction of positrons annihilate at vacancies located next to Sc atoms. A brief discussion and an additional figure have been added in the revised manuscript, please see page 12 and Supplementary Fig. 13.

Vacancies in (Cu, Sc, vacancy)-rich atomic complexes (*i.e.*, Sc-clusters) act as annihilation sites for positron annihilation. Based on the experimental results in Fig. 1f and Supplementary Table 1, the measured positron annihilation lifetime of AlCuSc-C alloy is mainly associated with the positron annihilation in bulk monovacancies and bulk divacancies. During cryogenic HPT of Al-Cu-Sc alloy, the diffusivity of deformation-induced vacancies was significantly suppressed by the cryogenic temperature, and the formation of (Cu, Sc, vacancy)-rich atomic complexes could further stabilize the vacancies. It is reasonable to assume that a large part of bulk monovacancies and bulk divacancies exist in the (Cu, Sc, vacancy)-rich atomic complexes. Thus, the positron annihilation lifetime in bulk monovacancies and bulk divacancies of the AlCuSc-C alloy arises mainly from the positron annihilation lifetime in (Cu, Sc, vacancy)-rich atomic complexes. Relevant discussions have been simply mentioned in the revised manuscript, please see page 13.

A-Figure 3 **a** CDB spectra for pure Al, pure Cu, pure Sc and the AlCuSc-C alloy. **b** CDB ratio curves for pure Cu reduced to 20% in amplitude, pure Sc, and the AlCuSc-C alloy. (P_L is the longitudinal component of the positron-electron momentum along the direction of the γ -ray emission)

Question 8. Page 6: Why is the lifetime in NC Al-Cu generally that high? What positron sinks are active there?

Response: Thanks for the reviewer's professional question. The annihilation of positrons is related to different vacancy types, *e.g.*, bulk monovacancy, and vacancies associated with dislocations. Vacancies can form through collision of moving dislocations and the vacancies can also diffuse into the dislocation core⁷, which are so called vacancies associated with dislocations. It has been generally accepted that the positron lifetime of vacancies associated with dislocations and of bulk monovacancies in Al is ~ 220 ps and ~ 245 ps in Al^{7,16}, respectively. Typically, the positron lifetime of room-temperature HPT-processed Al alloys falls within the range of ~ 215 ps - ~ 238 ps^{16,17}, which covers the measured lifetime in the first three alloys aforementioned. The measured lifetime of NC Al-Cu alloys in present work was ~ 228 ps in the AlCu-R alloy, ~ 226 ps in the AlCu-C alloy, ~ 236 ps in the AlCuSc-R alloy, and ~ 258 ps in the AlCuSc-C alloy. In the first three alloys mentioned above, owing to the presence of high-density dislocations (see Supplementary Fig. 3c), a part of the HPT-induced vacancies were trapped by the dislocations, and the remanent vacancies existed as bulk monovacancies. The measured lifetime of the three alloys is mainly due to the positron

annihilation in vacancies associated with dislocations and bulk monovacancy, see Fig. 1f and Supplementary Table 1. The AlCuSc-C alloy, however, exhibits a lifetime larger than that for bulk monovacancy. Since the positron lifetime increases with the size of vacancy-type defect^{6,18}, it indicates that a large fraction of defects containing more than one vacancy have formed in the AlCuSc-C alloy. As the positron lifetime of a divacancy is ~ 273 ps in Al⁷, the measured lifetime in the AlCuSc-C alloy should be dominated by the positron annihilation in both bulk monovacancies and divacancies. Referring to Fig. 1f and Supplementary Table 1, the fraction of monovacancies is about 53.5%, and the divacancies is about 46.5%. It should be mentioned that the formation of (Cu, Sc, vacancy)-rich atomic complexes in the AlCuSc-C alloy can prevent the vacancies from trapping at dislocation. Similar phenomenon has been also observed in Mg-bearing Al-Cu-Mg alloy, where the Cu and Mg atoms can stabilize free volume to form vacancy complex¹⁹.

Question 9. Page 9-10: I cannot see the formation of an enrichment up to 5% Cu in GP-zones after aging from Fig 4.

Response: Thanks for the reviewer's valuable suggestion. Plate-shaped Cu atom clusters with an enrichment up to 5.0 at.% are marked by dashed ellipse in Fig. 4d in the revised manuscript.

Question 10. Page 11: Fig 4c looks like typical L12 Al3Sc clusters. It is not clear to me why the FFT is not interpreted like that.

Response: Thanks for the reviewer's professional question. The three-dimensional structures of L1₀ and L1₂ chemical ordered structure in the real space are exhibited in **A-Figure 4 a** and **b**. A typical atomic-resolution HAADF-STEM image is presented in **A-Figure 4 c** to show the (Cu, Sc, vacancy)-rich atomic complex. It can be clearly found that some atomic columns are brighter than others in the **A-Figure 4 c**. To see intensity distribution of the brighter atomic columns, the intensity profiles of atomic columns on a (100) atomic plane (marked by yellow “[]” and yellow arrow) and on a (110) atomic plane (marked by red “[]” and red arrow) are

presented in **A-Figure 4 d** and **e**, respectively. No apparent alternated strong-weak intensity distribution of atomic columns can be found on the (100) and (110) plane. While in the $L1_0$ structure, along the [010] direction on the (100) plane, Al atomic columns and Sc atomic columns appear alternately. Similarly in the $L1_2$ structure of Al_3Sc , Al atomic columns and Sc atomic columns appear alternately along the [010] direction on the (100) plane and on an individual (110) plane. Besides, inverse FFT image of the (Cu, Sc, vacancy)-rich atomic complex in Supplementary Fig. 11b also clearly reveals no apparent alternated Al and Sc/Cu atomic columns. These comparisons demonstrate that some kind of short-range ordering exists in the (Cu, Sc, vacancy)-rich atomic complexes, which can not be categorized into the $L1_0$ or $L1_2$ structures.

A-Figure 4 **a** Three-dimensional structure of the $L1_0$ chemical ordered structure in the real space. **b** Three-dimensional structure of the $L1_2$ chemical ordered structure in the real space. **c** Atomic-resolution $\langle 001 \rangle_{Al}$ HAADF-STEM image showing the atomic complex. The intensity profiles of a (100) plane area (marked

with the yellow “[]”) and a (110) plane area (indicated by the red “[]”) along the directions indicated by the arrows in **c** are presented in **d** and **e**, respectively.

Question 11. Page 15: The representation of strain hardening rate in Fig. 5c is misleading. There is no physical valid curve through the different values. Why do the authors not use KM-plots derivate from Fig. 5a to calculate strain hardening rates?

Response: Thanks for the reviewer’s valuable suggestions. We agree that no physical valid curve exists through the different values in Fig. 5c. Thus, in the revised manuscript, we have re-plotted the figure. The reason for us not using the KM plots derivate from Fig. 5a to calculate the strain hardening rate is because the tensile testing of HPT-processed AlCuSc-C and AlCuSc-R alloys was based on small-sized samples while the tensile curves of coarse-grained Al-2.5Cu and Al-2.5Cu-0.3Sc alloys were based on standard tensile samples (gauge length of 30 mm). We are worrying about the comparability between the samples. For clarity, the KM plots derivate from Fig. 5a are exhibited in **A-Figure 5**. Two important parameters characteristic of strain hardening, θ_0 and β , can be extracted from the KM plots. θ_0 denotes the strain hardening rate at the beginning of plastic deformation. The characteristic of dynamic recovery, $\beta = -d\theta / d\sigma$, is defined as the slope marked by the dash-dotted lines in **A-Figure 5**. The AlCuSc-C alloy has a larger θ_0 and lower β when compared to the AlCuSc-R alloy, which evidences that the strain hardening rate is higher in the AlCuSc-C alloy. In order to further investigate and confirm the stronger strain hardening capability in the AlCuSc-C alloy, micropillar compressive tests were performed. Micropillar compression has been utilized to study the mechanical properties of materials at small length scales in a nominally homogeneous stress state²⁰ and has been successfully used to investigate the mechanical properties of nanocrystalline materials^{21,22}. In this work, the method proposed by Kiener and Minor²³ has been adopted to calculate the strain hardening rate of micropillar during compression^{24,25}, which is expressed as $\theta_{pillar} = \frac{\sigma_{5\%} - \sigma_{2\%}}{5\% - 2\%} n$. For single crystal samples in the micro- and sub-micrometer range, intermittent flow is usually observed in the compressive stress-strain curve, which is manifested by load drops or strain bursts during the test. Such

load drops or strain bursts are identified as dislocation avalanches using statistical methods on experimental data^{26,27} as well as simulation results²⁸. The KM plots are commonly obtained by plotting the instantaneous strain hardening rate as a function of $\sigma - \sigma_y$. Thus, for the stress-strain curves with intermittent flow (see Supplementary Fig. 15), the KM plots are not suitable for quantitatively describing the strain hardening. For this reason, the strain hardening rate (Fig. 5c) derived from Fig. 5b is adopted in present work.

A-Figure 5 Instantaneous strain hardening rate θ vs flow stress increment $\sigma - \sigma_y$ (Kocks-Mecking plots) calculated from the tensile engineering stress-strain curves of the AlCuSc-R, AlCuSc-C, peak-aged coarse-grained Al-2.5Cu and Al-2.5Cu-0.3Sc alloys in Fig. 5a.

Question 12. Page 17: The statement “It is soundly proved that the stronger strain-hardening capability in the II-C alloy should be related to atomic complexes.”, is a bit too strong from my opinion. This is only discussed based on indirect conclusions.

Response: Thanks for the reviewer’s professional and valuable suggestion. We have changed the sentence as “it can be concluded that the stronger strain hardening capability achieved in the AlCuSc-C alloy should be related to the presence of atomic complexes.” in the revised manuscript. Atomic complexes with short-range ordering have been reported to impact the

mechanical properties significantly²⁹⁻³¹. It has been demonstrated that the interaction between atomic complexes and dislocations can improve the strain hardening ability. On one hand, once the atomic complex is sheared by a leading dislocation, the following ones tend to glide on the same slip plane because of lower slip resistance. This “glide plane softening” effect^{30,31} can efficiently restrict cross-slip by promoting planar slip. It is known that the restriction of cross-slip can enhance the strain hardening ability via suppressing the annihilation of screw dislocations and dynamic recovery³². On the other hand, the interaction between atomic complex and moving dislocation can slow down the dislocation line migrating by pinning the dislocation, which can increase the opportunities for dislocations to interact with each other, leading to dislocation tangles and reactions²⁹. This process can also enhance the strain hardening ability. Despite of these previous conclusions, it is improper to state “It is soundly proved that...” in this manuscript, because our results were indirect.

Question 13. SFig.8: The only difference visible to me in the NN is statistics. There is no clear difference.

Response: Thanks for the reviewer’s valuable comment. We agree that the measured NN distance curve has only a very minor, though systematic, shifting in comparison to the random one. It is difficult to conclude the atom clustering of Cu atoms. Thus, in the revised manuscript, we have deleted the sentence “maybe with some kind of short-range ordering of Cu atoms occurring”.

Reviewer #3 (Remarks to the Author):

The authors present a highly original and extensive study where they report and explain the improved thermal stability of a nanostructured Al-Cu-Sc alloy in the presence of a supersaturation of vacancies. Explanations for improved thermal stability include experimental measurements (Positron Annihilation Spectroscopy) and theoretical calculations (density functional theory calculations of vacancy pairs with Cu and Sc atoms). Connection

between the solute-vacancy complexes and the mechanical properties (e.g. strain hardening) is also well-made. The work is likely to be impactful.

The present reviewer suggests a few changes and clarifications:

1. The authors need to reconsider the many superlatives they have used throughout the manuscript. How are “ultrahigh vacancy densities” different from high vacancy densities? The use of “ultrahigh thermal stability” on page 7 is especially objectionable for an alloy microstructure that is more thermally stable than other nanostructured alloys but not very thermally stable in an absolute sense since many Al alloys will have significantly higher thermal stability than the alloys under consideration.

Response: Thanks for the reviewer’s professional and valuable suggestion. “Ultrahigh vacancy densities” is replaced by “high vacancy densities”. “Ultrahigh thermal stability” is replaced by “high thermal stability”. Revisions are made accordingly in the revised manuscript.

2. Page 2 – line 56 – replace low dimensional materials with low dimension material geometries since the authors are referring to APT needles and TEM foils instead of low dimensional materials such as graphene and nanoribbons.

Response: Thanks for the reviewer’s professional suggestion. Revision is made accordingly, please see page 2 in the revised manuscript.

3. Figure 1 caption – replace “gains” with “grains”

Response: The reviewer is so careful in reviewing the manuscript and we are truly sorry for the carelessness. Revision is made accordingly.

References

1 Ziegler, J. F., Ziegler, M. D. & Biersack, J. P. SRIM – the stopping and range of

- ions in matter. *Nucl. Instrum. Methods Phys. Res. B* **268**, 1818–1823 (2010).
- 2 Gushchina, N. V., Ovchinnikov, V., Mozharovsky, S. M. & Kaigorodova, L. I. Restoration of plasticity of cold-deformed aluminum alloy by short-term irradiation with accelerated Ar⁺ ions. *Surf. Coat. Technol.* **389**, 125504 (2020).
- 3 Williams, D. B. & Carter, C. B. *Transmission Electron Microscopy: A Textbook for Materials Science*. (Plenum Publishing Corporation, 1996).
- 4 Ipatova, I., Harrison, R. W., Donnelly, S. E., Rushton, M. & Jimenez-Melero, E. Void evolution in tungsten and tungsten-5wt.% tantalum under in-situ proton irradiation at 800 and 1000 ° C. *J. Nucl. Mater.* **526**, 151730 (2019).
- 5 Dupasquier, A., Kogel, G. & Somoza, A. Studies of light alloys by positron annihilation techniques. *Acta Mater.* **52**, 4707–4726 (2004).
- 6 Cizek, J. *et al.* Thermal stability of ultrafine grained copper. *Physical Review B* **65**, 094106 (2002).
- 7 Su, L. H. *et al.* Study of vacancy-type defects by positron annihilation in ultrafine-grained aluminum severely deformed at room and cryogenic temperatures. *Acta Mater.* **60**, 4218–4228 (2012).
- 8 Nagai, Y. *et al.* Role of vacancy-solute complex in the initial rapid age hardening in an Al-Cu-Mg alloy. *Acta Mater.* **49**, 913–920 (2001).
- 9 Knipling, K. E., Seidman, D. N. & Dunand, D. C. Ambient- and high-temperature mechanical properties of isochronally aged Al-0.06Sc, Al-0.06Zr and Al-0.06Sc-0.06Zr (at.%) alloys. *Acta Mater.* **59**, 943–954 (2011).
- 10 Du, Y. *et al.* Diffusion coefficients of some solutes in fcc and liquid Al: critical evaluation and correlation. *Mater. Sci. Eng., A* **363**, 140–151 (2003).
- 11 Huang, Y., Robson, J. D. & Prangnell, P. B. The formation of nanograin structures and accelerated room-temperature theta precipitation in a severely deformed Al-4 wt.% Cu alloy. *Acta Mater.* **58**, 1643–1657 (2010).
- 12 Hu, T., Ma, K., Topping, T. D., Schoenung, J. M. & Lavernia, E. J. Precipitation phenomena in an ultrafine-grained Al alloy. *Acta Mater.* **61**, 2163–2178 (2013).
- 13 Huang, J., Meyer, M. & Pontikis, V. Is pipe diffusion in metals vacancy controlled? A molecular dynamics study of an edge dislocation in copper. *Phys. Rev. Lett.* **63**, 628–631 (1989).
- 14 Zhao, Y. H., Liao, X. Z., Jin, Z., Valiev, R. Z. & Zhu, Y. T. Microstructures and mechanical properties of ultrafine grained 7075 Al alloy processed by ECAP and their evolutions during annealing. *Acta Mater.* **52**, 4589–4599 (2004).
- 15 Honma, T., Yanagita, S., Hono, K., Nagai, Y. & Hasegawa, M. Coincidence Doppler broadening and 3DAP study of the pre-precipitation stage of an Al-Li-Cu-Mg-Ag alloy. *Acta Mater.* **52**, 1997–2003 (2004).
- 16 Lechner, W., Puff, W., Mingler, B., Zehetbauer, M. J. & Würschum, R. Microstructure and vacancy-type defects in high-pressure torsion deformed Al-Cu-Mg-Mn alloy. *Scr. Mater.* **61**, 383–386 (2009).
- 17 Lechner, W., Puff, W., Wildeb, G. & Würschum, R. Vacancy-type defects in amorphous and nanocrystalline Al alloys: Variation with preparation route and processing. *Scr. Mater.* **62**, 439–442 (2010).
- 18 Liu, F. *et al.* Characterization of radiation defects in quasi-homogeneously damaged

- tungsten by PALS and TEM. *Nucl. Instrum. Methods Phys. Res.* **439**, 17-21 (2019).
- 19 Marceau, R. K. W., Sha, G., Ferragut, R., Dupasquier, A. & Ringer, S. P. Solute clustering in Al-Cu-Mg alloys during the early stages of elevated temperature ageing. *Acta Mater.* **58**, 4923-4939 (2010).
- 20 Greer, J. R. & Hosson, J. T. M. Plasticity in small-sized metallic systems: Intrinsic versus extrinsic size effect. *Prog. Mater. Sci.* **56**, 654-724 (2011).
- 21 Wu, G. *et al.* Hierarchical nanostructured aluminum alloy with ultrahigh strength and large plasticity. *Nat. Commun.* **10**, 5099 (2019).
- 22 Liu, C. *et al.* Massive interstitial solid solution alloys achieve near-theoretical strength. *Nat. Commun.* **13**, 1102 (2022).
- 23 Kiener, D. & Minor, A. M. Source-controlled yield and hardening of Cu(1 0 0) studied by in situ transmission electron microscopy. *Acta Mater.* **59**, 1328-1337 (2011).
- 24 Niu, R. M. *et al.* Mechanical properties and deformation behaviours of submicron-sized Cu-Al single crystals. *Acta Mater.* **223**, 117460 (2021).
- 25 Zhang, J. Y., Lei, S. Y., Liu, G., Niu, J. J. & Sun, J. Length scale-dependent deformation behavior of nanolayered Cu/Zr micropillars. *Acta Mater.* **60**, 1610-1622 (2012).
- 26 Ng, K. S. & Ngan, A. H. W. Stochastic nature of plasticity of aluminum micro-pillars. *Acta Mater.* **56**, 1712-1720 (2008).
- 27 Dimiduk, D. M., Woodward, C., Lesar, R. & Uchic, M. D. Scale-Free Intermittent Flow in Crystal Plasticity. *Science* **312**, 1188-1190 (2006).
- 28 Csikor, F. F., Motz, C., Weygand, D., Zaiser, M. & Zapperi, S. Dislocation Avalanches, Strain Bursts, and the Problem of Plastic Forming at the Micrometer Scale. *Science* **318**, 251-254 (2007).
- 29 Chen, X., Wang, Q., Cheng, Z., Zhu, M. & Ma, E. Direct observation of chemical short-range order in a medium-entropy alloy. *Nature* **592**, 712-716 (2021).
- 30 Zhang, R., Zhao, S., Ophus, C., Deng, Y. & Minor, A. M. Direct imaging of short-range order and its impact on deformation in Ti-6Al. *Sci. Adv.* **5**, eaax2799 (2019).
- 31 Zhang, R., Zhao, S., Ding, J., Chong, Y. & Minor, A. M. Short-range order and its impact on the CrCoNi medium-entropy alloy. *Nature* **581**, 283-287 (2020).
- 32 Argon, A. *Strengthening Mechanisms in Crystal Plasticity*. (Oxford University Press on Demand, 2008).

REVIEWERS' COMMENTS

Reviewer #2 (Remarks to the Author):

The authors have really put a lot of effort into answering the questions raised and have improved the work considerably. Although I am not completely convinced of all the points they raise, I think it is difficult to improve this topic further, and there is not much room to go deeper into this experimentally difficult topic of characterising vacancies and solute vacancy complexes. Some speculation may remain, but this is on the cutting edge and I am convinced by the efforts made.